# A GNSS receiver positioning algorithm based on weighting the statistical properties of discontinuous signals at lower rotation speeds

Jiahui Gan[1,2,3], Peng Wu [1,2,3]*, Lu Feng[1,2,3], ZhiChun Dai[4], Rundong Li[1,2,3]

1 College of Electronic Communication and Electrical Engineering, Changsha University, Changsha, China, 2 Hunan Province Navigation and Attitude Measurement Integrated Application Engineering Technology Research Center, Changsha, China, 3 Changsha Low Earth Orbit Satellite Spatiotemporal Information Perception Technology Innovation Center, Changsha, China, 4 Hunan Satellite Navigation Information Technology Co., Ltd., Changsha, China

* wupeng@ccsu.edu.cn

## Abstract

During relatively low-speed rotation, the receiver captures the signal only when the antenna is turned in the direction of the satellite due to the long signal loss, a process in which the signal may be affected by noncontinuous reception. The conventional numerical preprocessing method for mitigating this is to reject the data with poor convergence as outliers. The conventional approach to this effect is to reject the poorly converged data as outliers, however, in non-continuous reception environments, simple rejection reduces the available satellite data and also affect the satellite geometric distribution, which further deteriorates the positioning results. In this paper, we propose a Global Navigation Satellite System (GNSS) receiver positioning algorithm based on weighting the statistical characteristics of discontinuous signals at lower rotational speeds. The method references the preorder available positioning results and employs a Kalman filtering algorithm designed based on independent model identification of observation weights to fully utilize the observation data of each satellite and improve the positioning accuracy. According to experimental data, the horizontal positioning error is reduced from 37.16 m with the original positioning algorithm to 28.31 m with the proposed method at 1/18 r/s, as an example.

## 1. Introduction

The positioning of rotating carriers remains a key area of research in satellite navigation technology. Positioning of rotating carriers remains a focus of research in satellite navigation technology. For example, rockets, artillery shells and missiles flying in the air rarely have the problem of satellite signal blocking. However, when these carriers spin to control their attitude during flight, the reception of satellite navigation signals becomes discontinuous, which affects the continuity and accuracy of their positioning. This phenomenon is more obvious when the carrier's rotational speed is low.

**Data availability statement:** All relevant data are within the paper and its Supporting Information files.

**Funding:** This paper was supported by major projects of the Changsha Science and Technology Bureau in 2020 (kq2011001), key research and development projects of the Hunan Provincial Department of Science and Technology in 2022 (2022GK2026), the Hunan Natural Science Foundation Project (2022JJ30636), and the Excellent Youth Program of the Scientific Research Program of the Department of Education of Hunan Province (22B0838), and the science and technology plan project of Hunan Provincial Department of Natural Resources (2023-78), the Aid Program for Science and Technology Innovative Research Team in Higher Educational Institutions of Hunan Province, and the Open Fund of the Xi'an Key Laboratory of Integrated Transport Big Data and Intelligent Control (Chang'an University) (Project No.: 300102343515).

**Competing interests:** The authors have declared that no competing interests exist.

The main solution to this type of problem is currently still to use positioning algorithms for satellite navigation to optimize under single-antenna conditions. In the traditional single-antenna solution, although the accurate continuous attitude estimation algorithm based on a single antenna can improve the attitude accuracy, the improved accuracy is limited. Under a certain rotational speed, the receiver can rely on its own baseband loop to maintain the signal tracking state and complete the localization solution, which is similar to a conventional receiver. However, at relatively low rotational speeds, the receiver can rely only on the limited time signal when the antenna is turned to the satellite direction to complete the signal tracking and convergence, which may be impacted by short signal reception times, i.e., noncontinuous reception, which may not be enough time to complete the capture. Furthermore, the signal may be lost again before observation data can converge. For this issue, the conventional numerical preprocessing method can be used to address the effects of poorly converged data by eliminating them as outliers. The traditional method in this non-continuous reception environment, which itself has less available satellite observation data, simply culls the data, which can easily cause the number of available satellite observation data to further decrease, plus this sudden decrease in the number of satellites can also affect the satellite geometric distribution, i.e., the Position Dilution of Precision (PDOP) value, causing further deterioration in the positioning results. PDOP is the square root of the sum of squared errors in latitude, longitude and elevation.

Carriers such as rockets and artillery shells, which use in-flight spin in the air, are protected from terrain factors to ensure attitude stabilization and eliminate the effects of mating errors on flight attitude. To achieve navigation capability, antennas must be added to or modified on the surface of these carriers, such as short, circularly polarized loop patch antennas in the 3–5 GHz band [1]. In the traditional single-antenna solution, although the accurate continuous attitude estimation algorithm based on a single antenna can improve the attitude accuracy, the improved accuracy is limited [2]. When the antenna is mounted on top of the rotating carrier, changes in the carrier's attitude can cause obstructions that limit the reception range of satellite signals [3,4]. However, if the antenna is mounted on the side of the rotating carrier and follows the carrier's center axis within the cross-section of the circular movement, the antenna signal range expands beyond the original fixed reception angle range. For instance, the antenna can receive a range of 120° to 160° and even up to 360°. The captured signals' angular range is inconsistent for a compact broadband circularly polarized (CP) antenna or other antenna types, such as common patch antennas [5–8]. In [9], for example, the effect of a GNSS receiving patch antenna radiation direction map on high-speed localization and mitigation techniques was discussed. Another application of a high-gain slit patch antenna was proposed in [10,11]. However, the rotation of the carrier causes the signal received by the receiving antenna to be noncontinuous, so ensuring stable tracking of the signal and accuracy of the observation is difficult [12]. While it is possible to ensure stable signal tracking and observational volume accuracy by equipping antennas with measurement-grade or higher quality antennas, it is important to consider the additional resource

consumption required for this approach [13,14]. Overseas scholars have researched this topic extensively for many years, primarily by implementing a rotary tracking loop to address the impact of rotation on the received signal. Compared to dual antennas, single antennas have lower installation requirements, are simpler to calculate and are used in more scenarios; furthermore, installing dual antennas on small-sized carriers is sometimes difficult. Among the available options, single antennas have a broader range of application scenarios. However, when a single antenna is used for measurement purposes, three main types of errors can occur. 1) Errors in speed measurement and magnitude; 2) Antenna placement and phase center bias; 3) Environmental motion effects such as multipath and terrain-induced disturbances. [15].

Installing multiple antennas has been attempted due to the potential instability of signal reception at lower speeds when relying on a single antenna. For example, antennas are symmetrically mounted on both sides of the rotating carrier [16] to ensure that at least one of the antennas can receive the satellite signals during the rotation process and to ensure that the range of the received signals is extended [17,18]. The signal characteristics of multiantenna signals [19] reveal that it has been experimentally derived that four antennas ensure a strong signal from the satellite at any given moment [20]. Fenton, in contrast, utilized a dual-antenna experimental environment to synthesize the signals received by each of the dual antennas to achieve accurate localization [21]. In [22], a carrier phase localization method was proposed based on dual antennas. In [23], a GNSS attitude determination method for multiantenna platforms was proposed. All of these methods utilize multiple antennas to receive signals, thus enhancing the capture of discontinuous signals; however, more antennas do not inherently correlate to a better actual performance. The main shortcomings of these methods include the following: 1) when multiple antennas are used to receive signals, the signals between antennas are prone to interference; 2) the inconsistency of hardware delay must be calibrated and compensated before use, which makes the algorithms require much more work in practice; 3) the noise of multiple antennas receiving signals is greater than that of single-antenna signals, especially in the rotating conditions, where the signals are weaker than in the regular case; and 4) the processing schemes may differ, but the amount of computation generally increases as the number of antennas increases [24,25]. Literature [26] suggests that when multiple antennas are considered, it is important to minimize mutual coupling and correlation in order to improve performance. Literature [27] presents a comparative study of different methods for reducing antenna mutual coupling, particularly in cases where the antenna spacing is less than about half of the wavelength. This coupling can distort the performance of antenna arrays, and therefore it is important to reduce it. This illustrates that the single-antenna signal coverage capability is limited and that multiple antennas have various disadvantages, such as the effect of merging noise and the effect of different antenna hardware delay differences. Other factors, such as product cost, volume, and reliability, must also be considered. As a result, one should try using fewer antennas to satisfy environmental needs.

For the localization solution under low rotational speed studied in this paper, the receiver can rely on capturing the signal only when the antenna is facing the direction of the satellite to recomplete the signal tracking and convergence, and the optimal coverage angle range of a single antenna is approximately 120°. Therefore, various auxiliary methods [28] can be utilized to further improve the positioning accuracy, such as one-way fuzzy time information [29], barometric altimetry [30], and approximate position estimation of visible satellites for assisted positioning [31]. However, considering the existence of signal anomalies [32] in the captured discontinuous signals, i.e., the captured satellites have not yet converged, their weights must be weakened [33] to ensure the accuracy of the localization solution.

In this study, the low rotational speed environment is examined, and related research directions are referenced. The commonly used patch antenna has an actual signal coverage angle of approximately 160°. By mounting two symmetrically placed antennas, the satellite signals can be captured in all directions. This approach has the potential to reduce the number of antennas needed to address the challenges of previous approaches while maintaining sufficient signal coverage in low-speed environments. In this non-continuous reception environment, where there are inherently fewer available satellite observations, the use of the weighted approach described in this paper not only improves the problem of positioning accuracy, but also improves the satellite geometric distribution. According to experimental data, the horizontal

positioning error is reduced from 37.16 m with the original positioning algorithm to 28.31 m with the proposed method at 1/18 r/s, as an example. Refer to the Experimental verification section for the improvement of the satellite geometric distribution.

This thesis is a GNSS receiver positioning algorithm based on the statistical characteristics of discontinuous signals at weighted lower rotation speeds, so this thesis first describes the signal characteristics of low rotation speed GNSS signals in the case of a single antenna, and then weights the standard Kalman filtering algorithm according to the signal characteristics, so that the influence of abnormal satellite signals on the positioning results will be reduced, i.e., the weight will be reduced, so as to achieve improvements in the positioning results and satellite geometric distribution. Finally, the method of this paper is experimentally verified, and the improvement of positioning results is reflected by the horizontal error and elevation error, and the improvement of satellite geometric distribution is reflected by the number of satellite receipts and PDOP value.

## 2. Laws of satellite navigation received signals during low-speed rotation

Since the receiver can rely on capturing the satellite signal only for the finite amount of time when the antenna is oriented toward the satellite, the duration of the signal can be used as a condition for modeling the observables in this paper. To illustrate the laws governing the reception of signals by antennas, we must discuss two directions: the improvement in the signal reception range due to antenna rotation and the new issue arising in signal reception due to antenna rotation.

### 2.1. Antenna rotation for signal reception range improvement

Mounting the antenna on top of a rotating carrier results in a significantly limited signal capture range, i.e., small pitch angles.

For comparison purposes, the carrier's vertical liftoff attitude is used as the comparison condition. As shown in Fig 1a, point is the position of the receiver rotating ca rrier, and $\alpha_i$ is the pitch angle of the satellite, where *i* corresponds to the

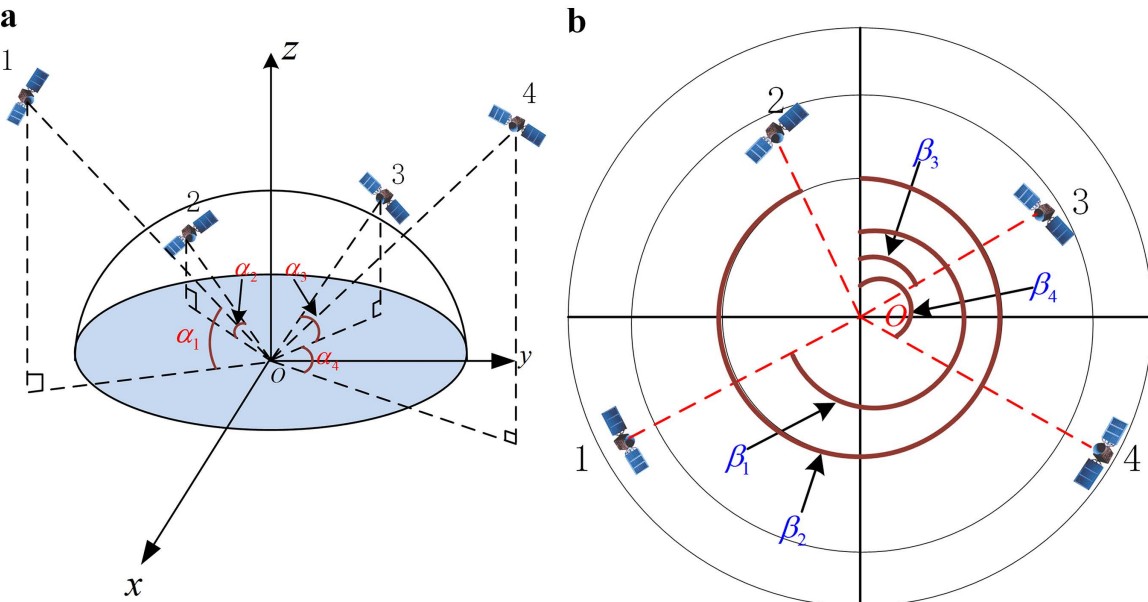

**Fig 1. Schematic diagram of the relationship between rotating carriers and satellite positions.** (a) Diagram of the antenna mounted on top of the carrier. (b) Antenna mounted on the side of the carrier.

satellite number. When the antenna is mounted on the top of the rotating carrier, the capture range of the signal exhibits a relatively large pitch angle. The satellites that can be captured in Fig 1a are satellites No. 1 and No. 4. However, given a relatively low pitch angle, for example, after the carrier has traveled to a certain altitude, the relative altitude and pitch angle of the satellite decrease. Satellites 2 and 3, as shown in the figure, are out of the capture range. As a result, the signal is not captured, and the number of usable stars used for positioning is reduced. However, when the antenna is mounted on the side of the rotating carrier, the capture range of the antenna increases as the rotating carrier rotates. As shown in Fig 1b, point O is still the position of the rotating carrier of the receiver, $\beta_i$ is the orientation angle of the satellite, which is viewed by using the perspective of looking down on the vertical lift-off carrier, and the satellite falls within different circular ranges representing different pitch angles. Stars 1 and 4 have larger pitch angles, and stars 2 and 3 have smaller pitch angles. When the antenna is mounted on the side of the rotating carrier, it will capture all satellites in the figure. However, there exists a special position of satellites, for example, near the Z direction of the coordinate axis in Fig 1a, that can always be received by the antenna mounted on the top of the coverage; in contrast, this position cannot be easily captured when the antenna is mounted on the side of the case, but in the case of Fig 1a, there is a possibility that the antenna mounted on the top of the case can never capture the signals. Of course, this is a rather extreme and rare case, and overall, the antenna is appropriately mounted on the side and rotated along with the carrier to obtain a greater signal reception range.

## 2.2. Emerging issues in signal reception due to antenna rotation

To enhance the signal reception range, a dual-antenna reception scheme is used in which the dual antennas are mounted symmetrically on both sides of the rotating carrier. This scheme is capable of doubling the antenna reception signal time, which is approximately one-third of the period of the rotating carrier before the dual antennas are mounted and approximately two-thirds of the period of the window after the dual antennas are mounted.

In Fig 2a, with a single antenna for localization, the optimal range of the antenna reception at each moment is $\gamma$, which has a value of approximately 120°, but the patch antenna is able to reach 160° in practice. Given a rotating carrier for rotation, the range of the signal capture covers all satellites from 1 to 6 in one cycle. In the initial state, the satellites within

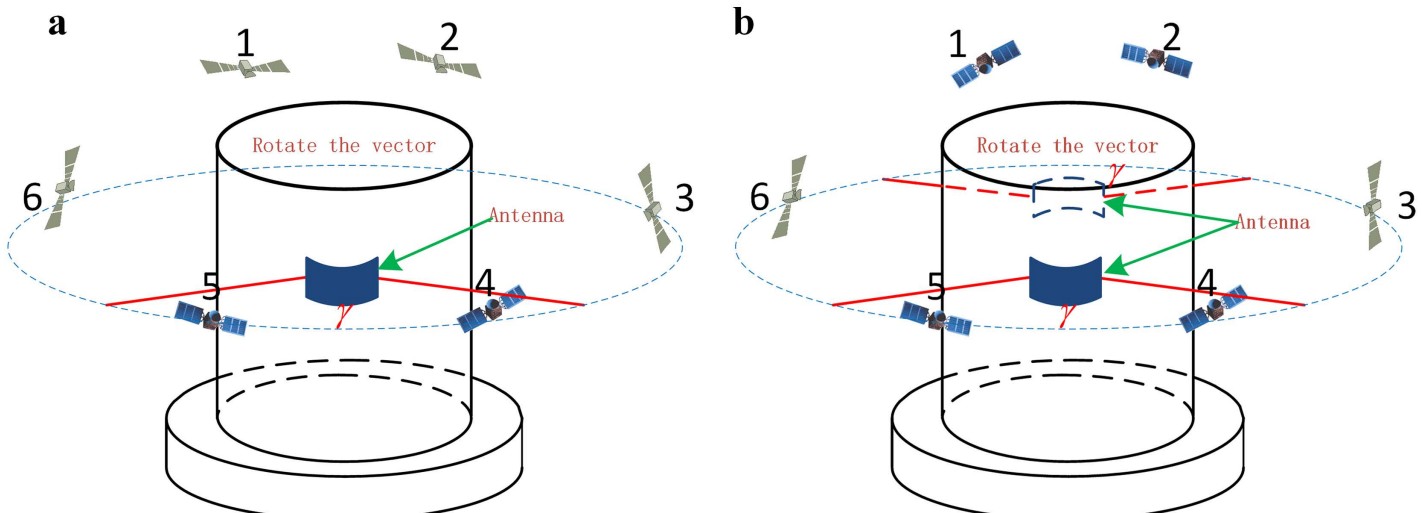

**Fig 2. Schematic of the capture range of a rotating carrier antenna.** (a) Schematic of the capture range of a single antenna on a rotating carrier. (b) Schematic of the capture range of the dual antennas of the rotating carrier.

the capture range are 4 and 5, corresponding to the blue icon satellites. When the antenna is next oriented toward satellites 4 and 5 and the rotation period of the carrier is long, satellites 4 and 5 cannot maintain the stability of the satellite signal by relying on their own loops. If the antenna is unable to orient toward satellites 4 and 5 before the satellite signal is lost, the extended rotation period leads to signal loss and ultimately abnormal positioning results. At this point, the antenna receives signals for 1/3 of the period. Under low rotation speed conditions, increasing the antenna signal receiving time is the key to improving the accuracy of the positioning solutions.

According to Fig 2b, by using the dual-antenna positioning and solution technique, the detection range at any given moment is $2\gamma$, which is 240°. Initially, the dual antenna captures satellites 4 and 5 and satellites 1 and 2, as indicated by the blue icons. After the rotating carrier has rotated 60°, the capturing satellites are still able to capture satellites 4, 5, 1 and 2, increasing the antenna signal reception time. With dual antennas, the optimum coverage of approximately 240° satellites can be maintained at all times so that a certain number of satellites are always captured during the rotation of the carrier, reducing the occurrence of satellite signal loss.

## 2.3. The effect of antenna size on signal reception at low speeds

Each antenna is designed to have a reception range of 120°, with the ability to receive satellite signals for 1/3 of the time in each cycle. Under experimental conditions using two antennas, the receiver can maintain a constant signal reception range of approximately 2/3 of the cycle time, resulting in an antenna signal reception time of approximately 2/3 of the cycle time. The antenna signal reception times at different rotation speeds are shown in Table 1.

As shown in the table, when a single antenna solution is used, the signal is in an undetectable state for a longer period. For example, when the speed drops to 1/18 r/s, the signal is interrupted for 12 s. In this case, maintaining the tracking state by relying solely on the receiver signal processing baseband loop is difficult. Even if it can be maintained, meeting accuracy requirements with the observable data provided may also be difficult. The use of dual antennas has greatly improved this situation. The time that the receiver loop is unable to receive the signal has been halved, reducing the difficulty of loop tracking. Furthermore, even if the signal must be completely recaptured, more time can be allowed for the signal to converge and stabilize, resulting in better observation accuracy. Therefore, in theory, the use of a dual-antenna solution can effectively improve the signal reception capability and the accuracy of the output observations.

However, many other factors affect this process. For example, when the speed is 1/4 r/s, the signal interruption appears to be only 1.3 s, but experience shows that by adjusting parameters such as the baseband loop bandwidth, there is a certain chance of maintaining the signal without losing lock. The problem in practice, however, is that to maintain signal capture capability under such dynamic rotation, the receiver loop bandwidth often needs to be increased accordingly, making it difficult for this scheme to rely on the receiver to maintain tracking capability when the loss of lock time is relatively short. However, there are some favorable aspects. For example, the conventional patch antenna design generally has a reception range of 120°, but in practice, it can often reach 160° or even more, which also has some favorable effects on the actual reception conditions. Of course, there will be some degradation in the performance of signals received at the edges of the antenna's coverage area.

Table 1. Window time for satellite signals at different rotation speeds.

| Number of revolutions per minute (r/s) | Cyclicality (s) | Receive signal time per antenna during the cycle (s) | Time each antenna was unable to receive a signal during the cycle (s) | Time that signal could not be received per cycle using dual antennas (s) |
|---|---|---|---|---|
| 1/4 | 4 | 1.3 | 2.7 | 1.3 |
| 1/6 | 6 | 2 | 4 | 2 |
| 1/12 | 12 | 4 | 8 | 4 |
| 1/18 | 18 | 6 | 12 | 6 |

Therefore, based on the above analysis, we can confirm that this is a complex experimental situation, and fully analyzing the issues based solely on theory or ideal conditions is impossible. Actual data collection and analysis are needed to further establish the laws of observation under low-speed conditions.

## 2.4. Analysis of actual received signals at low rotational speeds

The above analysis theoretically demonstrates that dual antennas can provide twice the signal reception time and can provide better theoretical performance whether the receiver relies on the baseband loop to maintain tracking or recaptures the signal after losing lock. To analyze the collected data patterns, we conducted satellite measurements in a dual-antenna environment and studied the signal variation of each satellite. This included examining the pseudorange values of the satellites and the number of satellites received[90°_20240717110227_003.dat].

Fig 3a displays the pseudorange values of a specific satellite. The pseudorange values were subtracted before and after processing to aid in detecting outliers caused by rotation. The pseudorange values undergo fluctuations as the rotating carrier rotates. When the antenna is directed toward the satellite in the antenna orientation chart, it captures the satellite's signal steadily, and the pseudorange value remains consistent without experiencing any fluctuations. When the antenna rotates away from the direction of the satellite and another antenna points toward the same satellite, the signal is recaptured, and the pseudorange value of the satellite experiences significant fluctuations. During this time, the pseudorange value experiences sudden fluctuations. The magnitude of these fluctuations is attributable to the restart of the loop filter, making its amplitude unpredictable. After a period has elapsed for the observed values to converge, the pseudodistance values return to normal. Thus, normal data still account for the majority of the time. If any aberrant data can be identified and utilized appropriately, positioning accuracy can still be assured. The graph in Fig 3b illustrates the alterations in the number of stars available for localization, which correspond to the changes in pseudorange values. Whenever the pseudorange value fluctuates, the number of stars available for localization correspondingly increases or decreases.

Analyzing satellite navigation signals received during low-speed rotation reveals that the data are highly error prone. Even the Kalman filtering method fails to overcome the large fluctuations in the data. While Kalman filtering models can be used to mitigate the unreliability of satellite data, it is important to consider their constraints. For instance, setting the

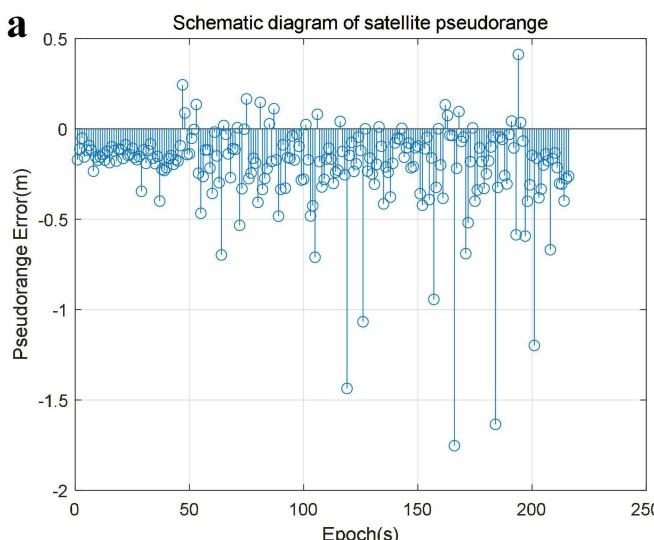
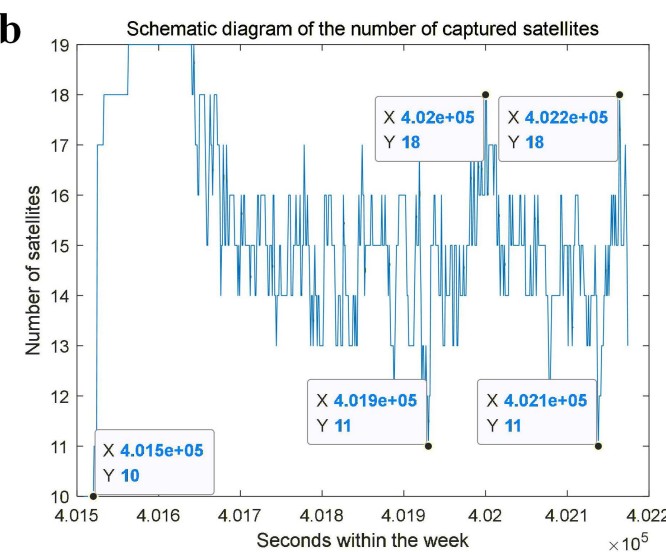

**Fig 3. Signal analysis diagram of a specific satellite.** (a) Schematic diagram depicting simulated observations of pseudorange values. (b) Schematic representation of the number of stars available for localization corresponding to pseudorange observations.

model to distrust highly variable data or using a stationary model with trajectory constraints may lead to filtering algorithms that do not adapt to the dynamics of certain receivers, such as remotely piloted UAVs with specific maneuvering characteristics. The appropriate model and constraints for avoiding the paradoxical problems commonly encountered with filters must be selected. In addition, relying solely on the model to reduce the weight of particularly large outlier values in the solving process can also impact the accuracy of the results. Instead, a more effective approach is to directly exclude these outliers from participating in the solving process through a simple judgment method.

Based on our previous theoretical analysis and measured data, the main solution to low-speed positioning accuracy degradation is the effective utilization of data. As a result, identifying the approximate error level of the data is crucial. When the accuracy level is high, directly using the data is best. Conversely, when accuracy is low, the data should not be used. When the accuracy is neither high nor low, a weighted solving method should be used to find a solution.

## 3. Kalman filter algorithm for determining observation weights using independent models

According to our previous analysis, we must address two issues. First, we must obtain the level of data error. To accomplish this, we can build a model based on the signal's historical state during stable tracking, allowing us to determine the current range of data accuracy. Second, we need to design a weight function to determine the data's weight.

The first issue can be addressed by utilizing intermittent, when-directed estimation methods for extrapolation, such as Chebyshev extrapolation, PE, and forgetting factor weighting. These approaches are suitable for this type of data. Considering the frequent loss of locks and interruptions in the data, as well as their relatively short duration, we have chosen the Chebyshev extrapolation algorithm.

For the second issue, there are numerous discussions of data preprocessing. When handling wildcard values that can be quite extensive and involve lengthy cut durations, certain pieces of data may become unusable. Even a less powerful weighting approach could still impact the outcomes. Therefore, we have established a straightforward guideline for eliminating the weakest data, utilizing the reliable data as usual and incorporating the intermediate data with reduced weight. This approach ensures the maximum availability of satellites and thus guarantees satisfactory PDOP accuracy, facilitating the execution of algorithms such as autonomous integrity check processing.

The method includes the following steps:

### 3.1. Establishment of the receiver equation of state for a GNSS-based positioning system

The Kalman filter is an optimal estimation method for the state of a discrete-time system that takes into account the estimation error, and its estimation process accounts for the correlation of several epochs before and after. The filtering process is chosen to describe the system state vector $\mathbf{x}_k$ at the kth epoch of an 11-dimensional system consisting of the position, velocity, and acceleration on the three coordinate components of the carrier (e.g., rockets, artillery shells) and the clock and frequency difference of the receiver[NavSolveKalman.c, NavSolveKalman.h]:

$$\mathbf{x}_k = \begin{bmatrix} x & v_x & a_x & y & v_y & a_y & z & v_z & a_z & \delta t_u & \delta f_u \end{bmatrix} \tag{1}$$

where the receiver's 3D position is represented by $x$, $y$, and $z$, their 3D velocity by $v_x$, $v_y$, and $v_z$, their 3D acceleration by $a_x$, $a_y$ and $a_z$, their clock difference by $\delta t_u$, and their frequency difference by $\delta f_u$.
Based on the state vector $\mathbf{x}_k$, the corresponding equation of state is:

$$\mathbf{x}_k = \mathbf{A}\mathbf{x}_{k-1} + \mathbf{B}u_{k-1} + w_{k-1} \tag{2}$$

where $\mathbf{A}$ is the state transfer matrix, $\mathbf{B}$ is the relationship matrix between the system inputs and the system state, $u_{k-1}$ represents the system inputs at the k-1st epoch, and $w_{k-1}$ represents the process noise vector at the k-1st epoch.

The state transfer matrix $\mathbf{A}$ is obtained as follows:

$$\mathbf{A} = \begin{bmatrix} 1 & T_S & \frac{T_S^2}{2} & & & & & & & & & \\ 0 & 1 & T_S & & & & & & & & & \\ 0 & 0 & 1 & & & & & & & & & \\ & & & 1 & T_S & \frac{T_S^2}{2} & & & & & & \\ & & & 0 & 1 & T_S & & & & & & \\ & & & 0 & 0 & 1 & & & & & & \\ & & & & & & 1 & T_S & \frac{T_S^2}{2} & & & \\ & & & & & & 0 & 1 & T_S & & & \\ & & & & & & 0 & 0 & 1 & & & \\ & & & & & & & & & 1 & T_S & \\ & & & & & & & & & 0 & 1 & \end{bmatrix} \tag{3}$$

where $T_S$ represents the discrete time interval of the state equation.

### 3.2. The receiver state is predicted based on the state transfer matrix $A$ in step 3.1

$\hat{\mathbf{x}}_{k-1}$ is assumed to be the optimal estimate of the Kalman filter for the system state $\mathbf{x}_{k-1}$ at the kth epoch. The a priori estimate $\hat{\mathbf{x}}_k^-$ of the Kalman filter for $\mathbf{x}_k$ at the kth epoch is determined.

$$\hat{\mathbf{x}}_k^- = \mathbf{A}\hat{\mathbf{x}}_{k-1} + w_k \tag{4}$$

where $w_k$ is the process noise vector at the kth epoch.
Then, the a priori estimation error $\mathbf{e}_k^-$ is:

$$\mathbf{e}_k^- = \mathbf{x}_k - \hat{\mathbf{x}}_k^- \tag{5}$$

In Kalman filtering, each a priori estimation error $\mathbf{e}_k^-$ must be followed by a mean-square error array $\mathbf{P}_k^-$ that measures the reliability of this a priori estimate $\hat{\mathbf{x}}_k^-$. The covariance of the process noise is added to the mean-square error array $\mathbf{P}_k^-$ of the a priori estimation error $\mathbf{e}_k^-$.

Let the covariance matrix $\mathbf{Q}$ of the process noise vector $w_k$ be:

$$\mathbf{Q} = E\left(w_k w_k^T\right) = \begin{bmatrix} \mathbf{Q}_x & & & & & & & & & \\ & \mathbf{Q}_v & & & & & & & & \\ & & \mathbf{Q}_a & & & & & & & \\ & & & \mathbf{Q}_x & & & & & & \\ & & & & \mathbf{Q}_v & & & & & \\ & & & & & \mathbf{Q}_a & & & & \\ & & & & & & \mathbf{Q}_x & & & \\ & & & & & & & \mathbf{Q}_v & & \\ & & & & & & & & \mathbf{Q}_a & \\ & & & & & & & & & \mathbf{Q}_t & \\ & & & & & & & & & & \mathbf{Q}_f \end{bmatrix} \tag{6}$$

Where $\mathbf{Q}_x$, $\mathbf{Q}_v$, $\mathbf{Q}_a$, $\mathbf{Q}_t$ and $\mathbf{Q}_f$ represent the process noise covariance of the receiver's position, velocity, acceleration, clock difference, and frequency difference, respectively. Therefore, the mean square error array $\mathbf{P}_k^-$ of the a priori estimation error $\mathbf{e}_k^-$ is:

$$\mathbf{P}_k^- = \mathbf{A}\mathbf{P}_{k-1}\mathbf{A}^T + \mathbf{Q} \tag{7}$$

where $\mathbf{P}_{k-1}$ denotes the state estimation mean square error array at the k-1st epoch.

### 3.3. Using the mean square error array $P_k^-$ of the a priori estimates derived from the prediction process of 3.2, the measurement variance is calculated

The individual state variables of the Kalman filter are observable, and their values can be directly or indirectly reflected in the system observations. Therefore, the observation vector $\mathbf{y}_k$ can be used to update the system state vector $\mathbf{v}_k$. The estimation can be accomplished based on the linear relationship that exists between the observation vector $y_k$ and the system state vector $\mathbf{x}_k$, which is as follows:

$$\mathbf{y}_k = \mathbf{C}\mathbf{x}_k + \mathbf{v}_k \tag{8}$$

$\mathbf{v}_k$ denotes the measurement noise vector, where the relationship matrix between the observation and the system state quantities is $\mathbf{C}$. If the position of a satellite is $(x^s, y^s, z^s)$ and the user's position is $(x,y,z)$ at a certain moment, the star-ground distance $L$ is:

$$L = \sqrt{(x^s - x)^2 + (y^s - y)^2 + (z^s - z)^2} \tag{9}$$

Then, the **C** matrix is:

$$\mathbf{C} = \begin{bmatrix} -\frac{x^s - x}{L} & 0 & 0 & -\frac{y^s - y}{L} & 0 & 0 & -\frac{z^s - z}{L} & 0 & 0 & 1 & 0 \end{bmatrix} \tag{10}$$

Based on Eq. (8) and the a priori estimate $\hat{\mathbf{x}}_k^-$ at the kth epoch, the difference between the observation vector $\mathbf{y}_k$ and the predicted value $\mathbf{C}\hat{\mathbf{x}}_k^-$ can be computed, called the observation vector residual $\Delta\mathbf{y}_k$:

$$\Delta\mathbf{y}_k = \mathbf{y}_k - \mathbf{C}\hat{\mathbf{x}}_k^- \tag{11}$$

Kalman filtering takes the linear combination of the a priori estimate $\hat{\mathbf{x}}_k^-$ with the residual $\Delta\mathbf{y}_k$ of the observation vector as the optimal estimate $\hat{\mathbf{x}}_k$ of the system state vector $\mathbf{x}_k$:

$$K_k = \mathbf{P}_k^-\mathbf{C}^T(\mathbf{C}\mathbf{P}_k^-\mathbf{C}^T + \mathbf{R})^{-1} \tag{12}$$

$$\hat{\mathbf{x}}_k = \hat{\mathbf{x}}_k^- + K_k\Delta\mathbf{y}_k \tag{13}$$

where $K_k$ is the filter gain, $\mathbf{R}$ is the covariance matrix of the measurement noise vector $\mathbf{v}_k$.
According to equation (5), the a posteriori estimation error $\mathbf{e}_k$ is:

$$\mathbf{e}_k = \mathbf{x}_k - \hat{\mathbf{x}}_k \tag{14}$$

Then, the mean square error array $\mathbf{P}_k$ of the a posteriori estimation error $\mathbf{e}_k$ is:

$$\mathbf{P}_k = E\left[\mathbf{e}_k\mathbf{e}_k^T\right] \tag{15}$$

The measurement variance $\sigma^2$ is calculated using the relationship matrix $\mathbf{C}$ between the observation vector $\mathbf{y}_k$ and the system state vector $\mathbf{x}_k$, the mean-square error array $\mathbf{P}_k$ for the a posteriori estimation error $\mathbf{e}_k$, and the mean-square error array $\mathbf{P}_k^-$ for the a priori estimation error $\mathbf{e}_k^-$:

$$\sigma^2 = \mathbf{C}\mathbf{P}_k^-\mathbf{C}^T + \mathbf{P}_k \tag{16}$$

### 3.4. Update the mean-square error array $\mathbf{P}_k$ of the a posteriori estimation error $\mathbf{e}_k$ in Step 3.3

**3.4.1. Using the Chebyshev polynomials to extrapolate the pseudodistance values and calculate the observation weight coefficients.** The pseudorange observation $p_k$ at the kth epoch is processed according to the Chebyshev polynomials to provide the weight settings for the final localization solution. According to the Chebyshev polynomials, the extrapolated pseudorange value $p(k)$ corresponding to the satellite at the kth epoch is[Chebshev.c, Chebshev.h]:

$$p(k) = \sum_{i=0}^{n} Q_{p,i} T_i(\tau_k) \tag{17}$$

where $Q_{p,i}$ is the Chebyshev coefficient corresponding to the extrapolated pseudorange value of the satellite, $n$ is the order of the Chebyshev polynomials, $i$ denotes the coefficient that sums the order of the Chebyshev polynomials, $i = 0, 1, \cdots n$, $T_i(\tau_k)$ is the Chebyshev polynomial, and $\tau_k$ is the interval of the independent variables of the Chebyshev polynomial, where $\tau_k$ is:

$$\tau_k = \frac{2(k-k_1)}{k_2 - k_1} - 1, k > k_1 > k_2 \tag{18}$$

where $k_1$ and $k_2$ are the epochs with known pseudorange, $k_1$ is the epoch corresponding to the first pseudorange observation $p_{k_1}$, and $k_2$ is the epoch corresponding to the last pseudorange observation $p_{k_2}$.

Then, the Chebyshev polynomial $T_i(\tau_k)$ is:

$$\begin{aligned} T_0(\tau_k) &= 1 \\ T_1(\tau_k) &= \tau \\ T_i &= 2\tau_k T_{i-1}(\tau_k) - T_{i-2}(\tau_k) \quad i..2 \end{aligned} \tag{19}$$

Assuming that the number of epochs involved in the extrapolation is $m\,(m \geq n)$, the Chebyshev polynomial matrix $\mathbf{T}$ consisting of Chebyshev polynomials $T_i(\tau_k)$ can be expressed as follows:

$$\mathbf{T} = \begin{bmatrix} T_0(\tau_1) \ T_1(\tau_1) \cdots T_n(\tau_1) \\ T_0(\tau_2) \ T_1(\tau_2) \cdots T_n(\tau_2) \\ \vdots \\ T_0(\tau_m) \ T_1(\tau_m) \cdots T_n(\tau_m) \end{bmatrix} \tag{20}$$

The Chebyshev coefficient matrix $\mathbf{Q}_p$ consisting of Chebyshev coefficients $Q_{p,i}$ and the pseudorange observation matrix $\mathbf{p}$ consisting of pseudorange observations $p_k$ are denoted as:

$$\mathbf{Q}_p = \begin{bmatrix} Q_{P,0} \\ Q_{P,2} \\ \vdots \\ Q_{P,n} \end{bmatrix}, \mathbf{p} = \begin{bmatrix} p_1 \\ p_2 \\ \vdots \\ p_m \end{bmatrix} \tag{21}$$

According to the principle of least squares, the Chebyshev coefficient matrix $\mathbf{Q}_p$ can be expressed as:

$$\mathbf{Q}_P = \left(\mathbf{T}^\mathsf{T}\mathbf{T}\right)^{-1}\left(c\mathbf{T}^\mathsf{T}\mathbf{p}\right) \tag{22}$$

**3.4.2. Set the equivalent weight function $\overline{\mathbf{p}}_k$ according to the extrapolated pseudodistance value $\mathbf{p}(k)$ of the Chebyshev polynomials of 3.4.1.** After iterative validation, the optimal parameters $a$ and $b$ are obtained under the statistical properties of the difference between the extrapolated pseudorange value $p(k)$ and the pseudorange observation $p_k$. Then, the equivalence weight function $\overline{p}_k$ is set according to the statistical properties, and the equivalence weight function $\overline{p}_k$ is as follows:

$$\overline{p}_k = \left\{ \begin{array}{ll} 1, & \left|p(k)\text{-}p_k\right| \le a \\ x_1/\left(\left|p(k)\text{-}p_k\right|\right)\text{-}x_2, & a < \left|p(k)\text{-}p_k\right| \le b \\ x_3/\left(\left|p(k)\text{-}p_k\right|\right), & \left|p(k)\text{-}p_k\right| > b \end{array} \right\} \tag{23}$$

The following results are derived from the weight function:

If $\left|p(k)\text{-}p_k\right| > b$, The data is considered highly anomalous, and the equivalence weight $\overline{p}_k$ is reduced as much as possible to minimize its impact on the localization solution.

If $\left|p(k)\text{-}p_k\right| \le a$, $\overline{p}_k = 1$, fluctuations in the data are considered to follow the white noise law and contribute normally to the solution.

If $a < \left|p(k)\text{-}p_k\right| \le b$, $0.1 \le \overline{p}_k < 1$, the data are used after a weak equivalence right $\overline{p}_k$.

Based on the equivalence weights, the mean square error array $\mathbf{P}_k{}'$ of the new a posteriori estimation error $\mathbf{e}_k$ is obtained as:

$$\mathbf{P}'_k = \mathbf{P}_k\overline{p}_k \tag{24}$$

The new measurement variance is:

$$\sigma^{2'} = \mathbf{C}\mathbf{P}_k^-\mathbf{C}^T + \mathbf{P}_k{}' \tag{25}$$

## 3.5. Updating the motion state of the receiver based on the described prediction process

After determining the measured variance value, the filter gain $K'_k$ can be further calculated, and the state estimate $\hat{\mathbf{x}}'_k$ can be measured and updated as shown in equations (26) and (27):

$$K'_k = \mathbf{P}_k^-\mathbf{C}^T(\sigma^{2'})^{-1} \tag{26}$$

$$\hat{\mathbf{x}}'_k = \hat{\mathbf{x}}_k^- + K'_k\Delta\mathbf{y}_k \tag{27}$$

## 4. Experimental verification

To verify the viability of the proposed localization method, a rotating carrier test-bed with dual antennas and an application environment are established, as illustrated in Fig 4.

Based on the experimental scenario diagram illustrated in Fig 4, the dual antenna positioning technique is employed for signal reception. Dual antennas are mounted on the side of the rotating carrier, and the antennas are symmetrically mounted on both sides, enabling at least one antenna to receive satellite signals during rotation. This expands the range of the received signals and makes capturing satellite signals easier. The delay caused by cable length is negligible in this

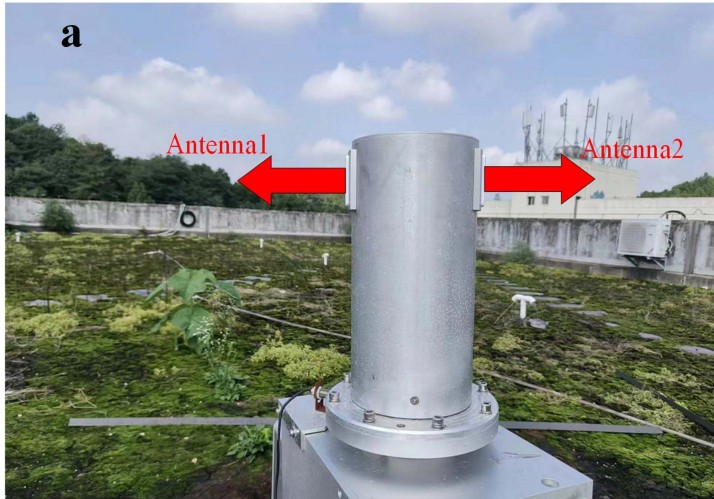
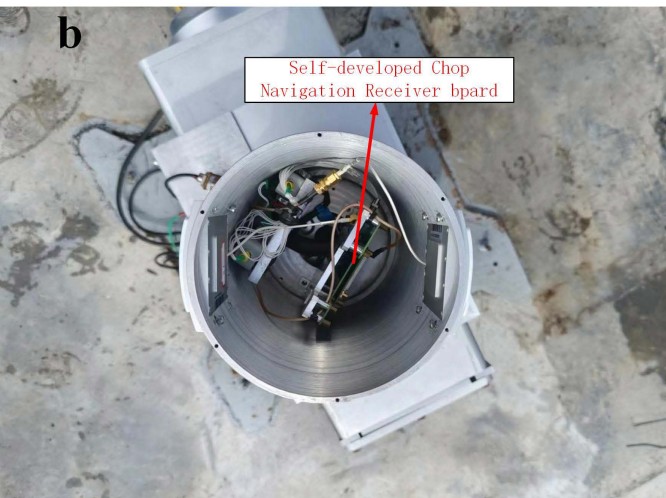
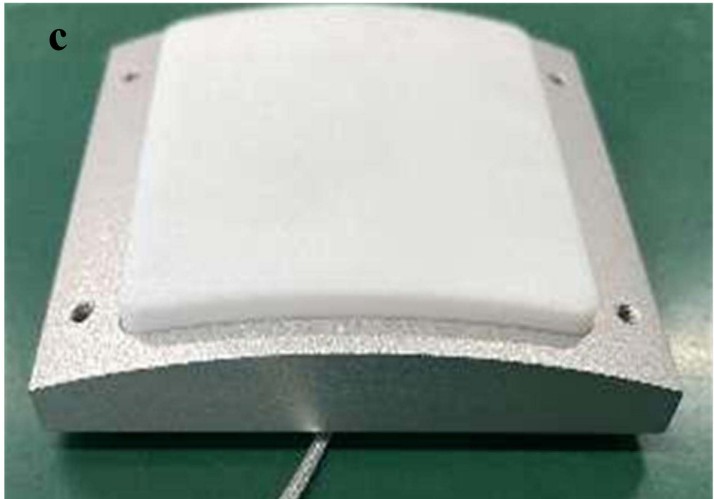
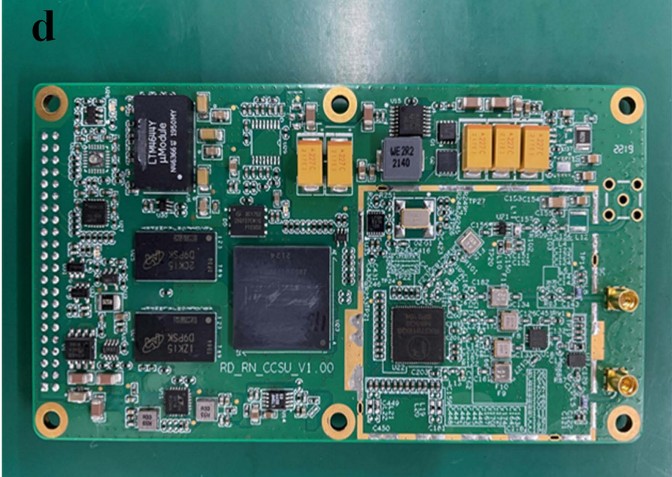

**Fig 4. Schematic diagram of the experimental scenario.** (a) Rotating carrier flat view. (b) Rotating carrier top view. (c) Antenna. (d) Self-developed chip navigation receiver board.

experimental scenario. The self-developed receiver board utilises the CNAV GEN1 chip, with a positioning measurement sampling rate of 1 Hz. The rotational speed accuracy of the rotating body is 0.01 r/s. Phase-locked loop order: 1st order Frequency-locked loop order: 1st order Phase-locked loop bandwidth: 24 Hz Frequency-locked loop bandwidth: 0 Hz.

The antenna model and antenna gain pattern are shown in the Fig 5 below[Antenna Gain Data.xls, Antenna.m].

The core parameters of the antenna are as follows: Maximum gain: 4.47 dB at (Phi＝−118.0°, Theta＝−2.0°). Gain range: −48.12 dB to 4.47 dB.

The above experimental scenarios were used to collect the low rotational speed rotation data of the rotating carrier, and the data was played back using the self-developed software shown in Fig. 6 to achieve the validation of the methodology of this paper.

Add fields to the existing structure to store prior error variances, weights and post-filtering pseudorange differences. This will enable the calculation of prior error variances. Use the $3\sigma$ rule to remove outliers from the pseudorange differences to enhance the accuracy of the variance estimation. Calculate the sample variance from the cleaned data to

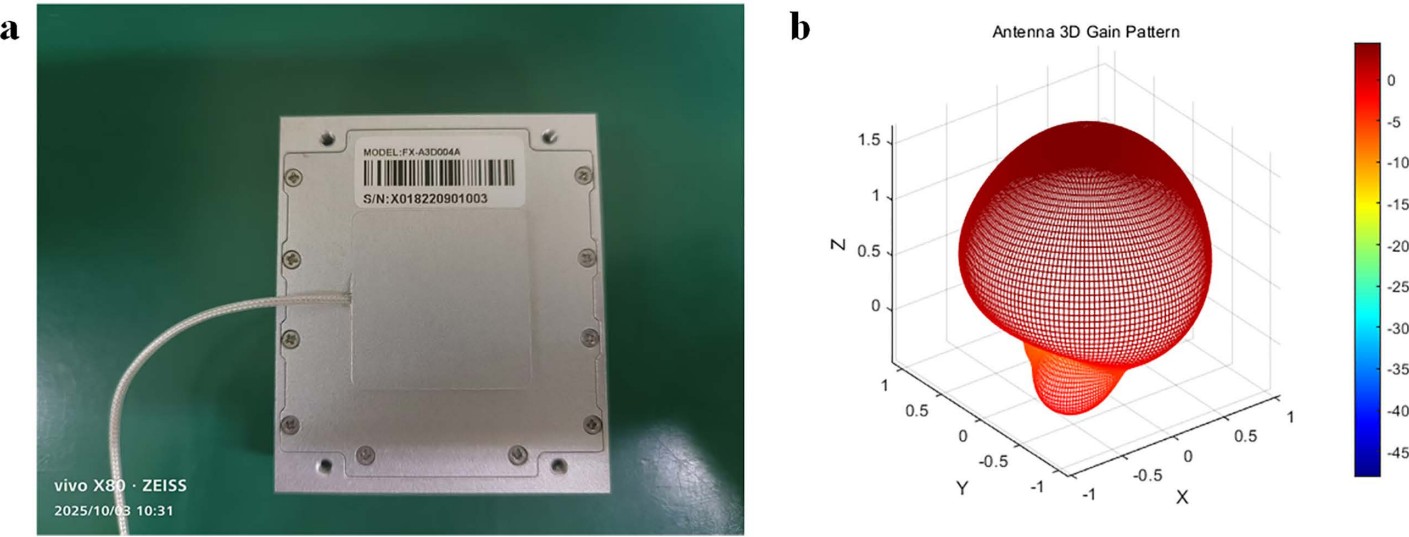

**Fig 5. Antenna Model and Antenna Gain Pattern.** (a) Antenna Model. (b) Antenna Gain Pattern.

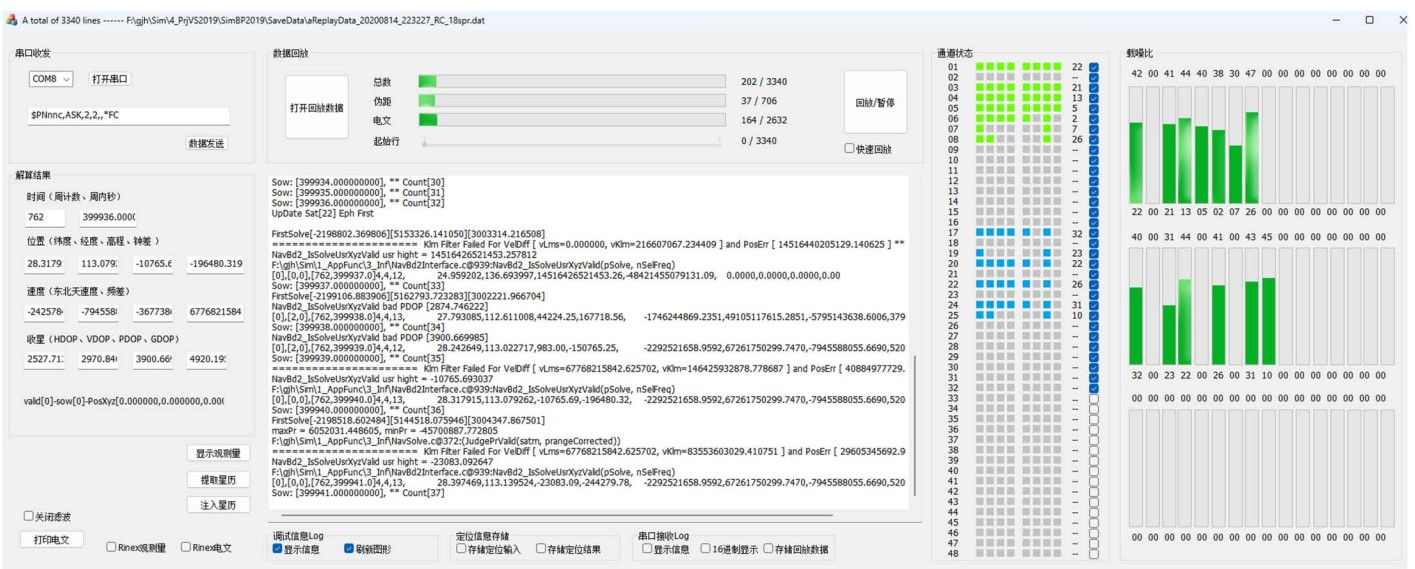

**Fig 6. Experimental verification software interface diagram.**

determine the prior error variance and implement weight fitting. Weights are computed using the inverse variance method (i.e., smaller errors yield larger weights). These weights are then applied to sliding window weighted filtering to demonstrate the weighting effect.

Fig 6 below illustrates the verification of the weight function for equation (23).

In Fig 7 presents the weighting function analysis results for Satellite 1, illustrating a clear comparison between dynamic weighting and fixed weighting (based on prior variance) in terms of pseudorange error filtering. The first subplot compares the original pseudorange error (blue curve), the result of dynamic weighted filtering (red curve), and the result of

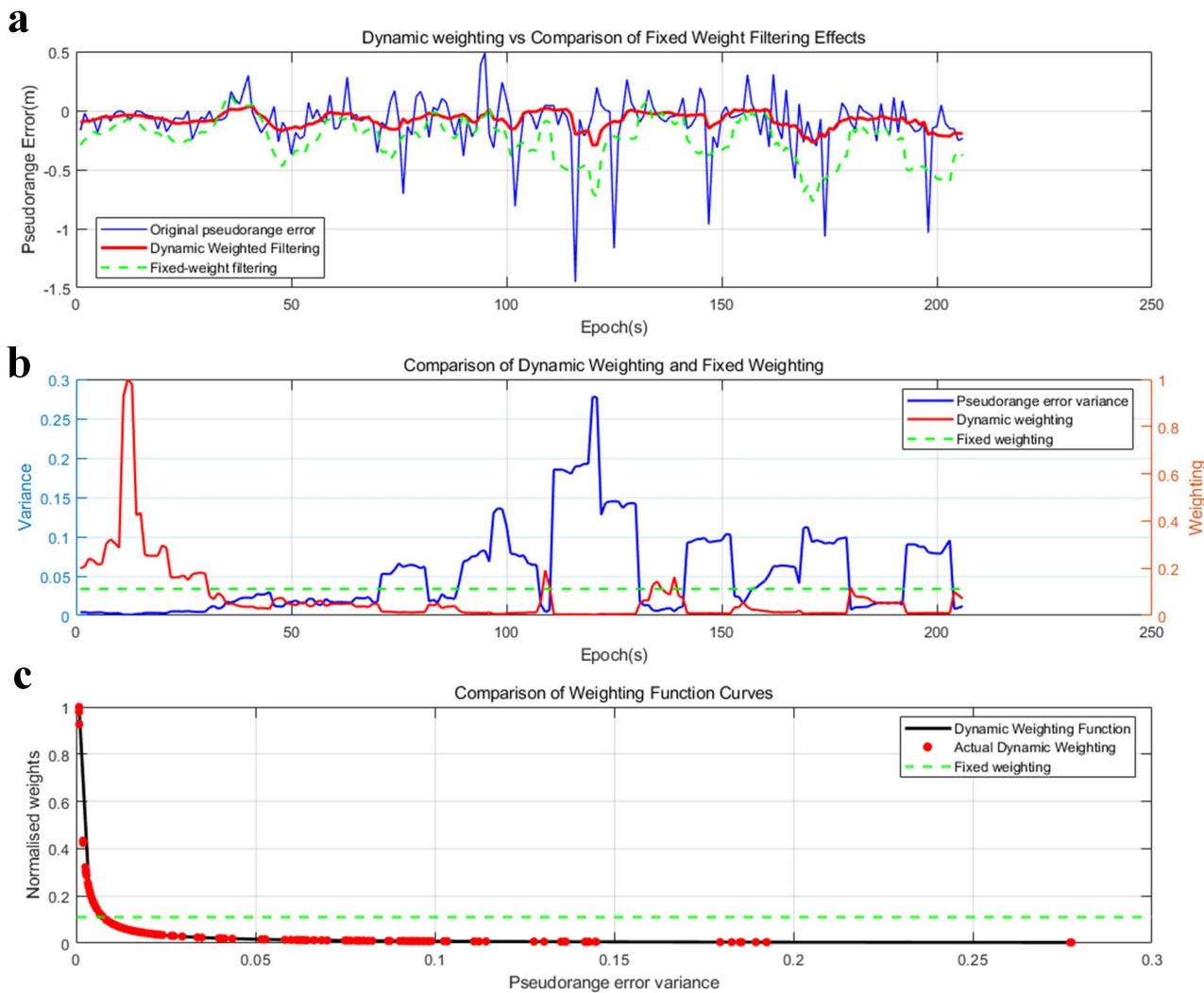

**Fig 7. Weighting Function Fitting Process Diagram.** (a) Dynamic weighting vs Comparison of Fixed Weight Filtering Effects. (b) Comparison of Dynamic Weighting and Fixed Weighting. (c) Comparison of Weighting Function Curves[90°_20240717110227_003.dat, WeightSet.m].

fixed-weight filtering (green dashed curve), showing that dynamic weighted filtering adapts more flexibly to the fluctuations of the original error, while fixed-weight filtering has a relatively stable smoothing effect. The second subplot displays the pseudorange error variance (blue curve, left axis), dynamic weighting (red curve, right axis), and fixed weighting (green dashed curve, right axis), indicating that dynamic weighting decreases significantly in regions with large pseudorange variance, reflecting its adaptive characteristic, whereas fixed weighting remains constant regardless of the variance fluctuations of individual epochs. The third subplot contrasts the weighting function curves, with the black curve representing the theoretical dynamic weighting function, the red dots denoting the actually calculated dynamic weights, and the green dashed line indicating the fixed-weight reference.

The above demonstrates the corresponding relationship between the variance of pseudorange error and the distribution weight over time, wherein the weight monotonically decreases as the variance increases. This reflects the inverse relationship inherent in the function design. The weight function can be fitted using the third subfigure above.

During the localization process, both the traditional filtering algorithm and the filtering algorithm that weighs the statistical characteristics of discontinuous signals at lower rotational speeds are applied. The experimental results of these two methods are then compared in terms of intraweek seconds. Intraweek seconds denote the number of seconds elapsed from the start of the week (Sunday at 00:00:00) to the present moment. Each GPS week comprises 604,800 seconds, hence weekly seconds range from 0 to 604,799. This metric is employed to denote time within a week with greater precision. Satellite Intraweek seconds = (UTC time − Satellite reference time + Time difference) ÷ 604,800 modulo. Time difference is Time difference between satellite time systems and UTC (Coordinated Universal Time). These results are presented in Fig 8[aNavData-New-high-4spr.csv, aNavData-New-high-12spr.csv, aNavData-New-high-18spr.csv, aNavData-Tradition-high-4spr.csv, aNavData-Tradition-high-12spr.csv, aNavData-Tradition-high-18spr.csv, aReplayData_RC_4spr.dat, aReplayData_RC_12spr.dat, aReplayData_RC_18spr.dat].

In Fig 8, the x-axis shows the time in seconds, and the y-axis shows the elevation in meters. This graph shows that the blue state is the outcome of elevation localization with the weighted solution method, while the red state is the result of elevation localization using the traditional filtering algorithm.

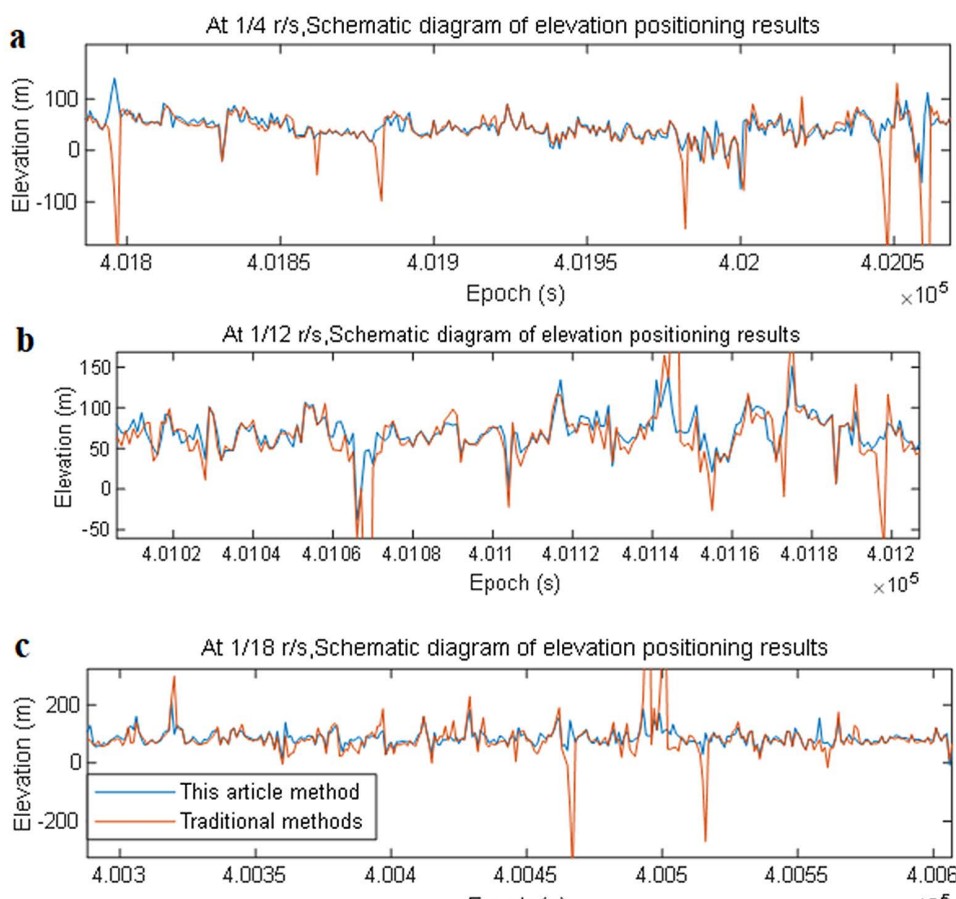

**Fig 8. Comparison of elevation localization results of this algorithm at varying rotational speeds.** Schematic diagram. (a) Comparison of elevation positioning results at 1/4 r/s. (b) Comparison of the elevation positioning results at 1/12 r/s. (c) Comparison of the elevation positioning results at 1/18 r/s.

During the experiment, the traditional filtering algorithm achieved results roughly within the acceptable error range when positioning was conducted at rotational speeds of 1/4 r/s, 1/12 r/s, and 1/18 r/s. However, there were instances of satellite observation abnormalities during carrier rotation, resulting in abnormal positioning results. This caused the positioning results to fluctuate, as shown by the red-state protrusions in the figure. The GNSS receiver positioning algorithm, which weights the statistical characteristics of the discontinuous signal using the method proposed in this paper, can enhance the traditional filtering algorithm by effectively allocating weights to visible positioning satellites and increasing the weights of the useful parts of visible satellite observation information. This method reduces the number of fluctuations in positioning results, ultimately improving the positioning accuracy.

Fig 9a, 9c, and 9e are schematic illustrations of the horizontal positioning error obtained through the conventional method for 1/4 r/s, 1/12 r/s, and 1/4 r/s, respectively. Correspondingly, Fig 9b, 9d, and 9f are schematic illustrations of the horizontal positioning errors of the proposed method for 1/4 r/s, 1/12 r/s, and 1/4 r/s, respectively. The error values are in meters. Comparing the two scatter plots reveals that the horizontal localization error results are more widely scattered and have a larger error margin when the traditional method is used. The horizontal positioning errors are denser when using the proposed method, resulting in more stable positioning outcomes. However, comparing the rotational speeds of 1/4 r/s, 1/12 r/s, and 1/18 r/s reveals that this algorithm's improved localization results are more pronounced at 1/18 r/s. Conversely, as shown by the 1/12 r/s and 1/4 r/s cases, the algorithm performs slightly worse at 1/18 r/s than at lower speeds. In other words, this algorithm's localization improvements are more obvious at lower rotational speeds.

As shown in Table 2, in this paper's method, the percentile values of the horizontal error at various rotational speeds are arranged from smallest to largest, with the P95 values being the values at the 95% position. At 1/4 r/s, the P95 value for the horizontal error of the proposed method is approximately 29.68 m, while the P95 value for the horizontal error of the traditional method is approximately 33.84 m. At a rotation speed of 1/6 r/s, the P95 value for the horizontal error of the proposed method is approximately 37.26 m, while the P95 value for the horizontal error of the traditional method is approximately 42.62 m. At a rotation speed of 1/12 r/s, the P95 value for the horizontal error of the proposed method is approximately 29.39 m, while the P95 value for the horizontal error of the traditional method is approximately 40.77 m. At a rotational speed of 1/18 r/s, the P95 value for the horizontal error of the proposed method is approximately 28.31 m, whereas the P95 value for the horizontal error of the traditional method is 37.16 m. Analyzing the error comparison at various rotational speeds reveals that the proposed method has an advantage and can offer a viable solution for localizing at low rotational speeds. All of these principles can enhance the accuracy of the original localization solution results. By comparing the error rates between the traditional and proposed methods at various rotational speeds, this research demonstrates the advantages of the latter method, which can improve the original localization solving results, especially at low rotational speeds.

Using Table 2, readers can compare horizontal positioning error percentile values more easily and see that the proposed method is superior to the traditional method.

To better demonstrate the methodology presented herein and enhance the representativeness of the algorithm, analyses were conducted for different satellite numbers. Two robust baselines—Huber-M and 3σ gating—were incorporated, alongside comparative plots of RMSE/MAE/P50/P68/P95 metrics. Across multiple runs, the confidence interval was set at 95%. The experimental results are illustrated in Figure 9 below. The data results are shown in Table 3 and Table 4 below.

Fig 10 presents a comprehensive comparison of the proposed dynamic weighting method against traditional fixed weighting and two GNSS-domain robust baselines (Huber-M estimation, 3σ gate-weighted filtering) for pseudorange error processing. The top subplot, focusing on filtering performance, shows the original pseudorange error (blue curve) with significant fluctuations and occasional gross errors, while the proposed method (red curve) achieves the smoothest suppression of both high-frequency noise and outliers—outperforming the rigid smoothing of fixed weighting (green dashed curve) and the residual fluctuations of the two robust baselines (pink dotted curve for Huber-M, cyan dotted curve for 3σ weighting). The middle subplot links pseudorange error variance (blue curve, left axis) to normalized weights (right axis),

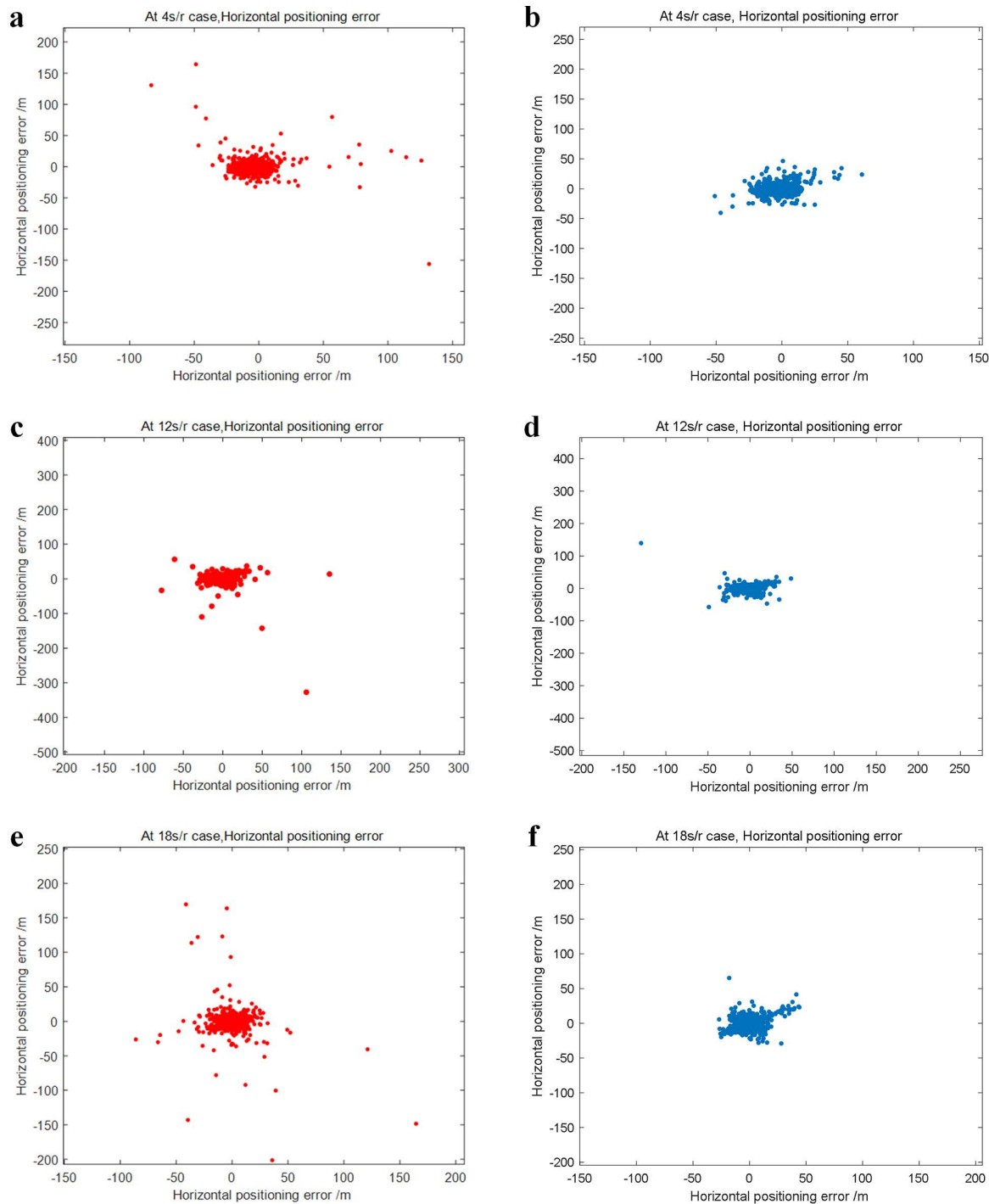

**Fig 9. Schematic comparison of the horizontal error between this algorithm and the conventional algorithm at various rotational speeds.**
(a) Schematic of the horizontal positioning error of the original method at 1/4 r/s. (b) Schematic of the horizontal localization error of the method in this paper at 1/4 r/s. (c) Schematic of the horizontal positioning error of the original method at 1/12 r/s. (d) Schematic of the horizontal localization error of the method in this paper at 1/12 r/s. (e) Schematic of the horizontal positioning error of the original method at 1/18 r/s. (f) Schematic of the horizontal localization error of the method in this paper at 1/18 r/s[aReplayData_RC_4spr.dat, aReplayData_RC_12spr.dat, aReplayData_RC_18spr.dat, func_XYZto-BLH.m, Nav_CalcBlhHoriErr.m, ShuiP.m].

**Table 2. Comparison table of percentile values for horizontal error at various speeds.**

| Number of revolutions per minute (r/s) | Percentile value of horizontal error of the traditional method (m) | Percentile value of horizontal error of the method in this paper (m) |
|---|---|---|
| 1/4 r/s | 33.84 | 29.68 |
| 1/12 r/s | 40.77 | 29.39 |
| 1/18 r/s | 37.16 | 28.31 |

**Table 3. Performance Comparison of Proposed Method vs Traditional/Robust Baseline.**

| Satellite | Indicator | Primitive | Fixed weighting | Huber-M | 3σ gate | Methodology of this paper |
|---|---|---|---|---|---|---|
| Satellite 1 | RMSE(m) | 0.2345 | 0.2967 | 0.1569 | 0.1452 | 0.1476 |
| Satellite 1 | MAE(m) | 0.1421 | 0.2328 | 0.1000 | 0.1021 | 0.1017 |
| Satellite 1 | P50(m) | 0.0699 | 0.1611 | 0.0575 | 0.0680 | 0.0644 |
| Satellite 1 | P68(m) | 0.1422 | 0.2811 | 0.0990 | 0.1166 | 0.1219 |
| Satellite 1 | P95(m) | 0.4810 | 0.6347 | 0.3590 | 0.3106 | 0.3460 |
| Satellite 2 | RMSE(m) | 0.1465 | 0.2451 | 0.1202 | 0.1157 | 0.1150 |
| Satellite 2 | MAE(m) | 0.0917 | 0.1918 | 0.0784 | 0.0811 | 0.0778 |
| Satellite 2 | P50(m) | 0.0549 | 0.1541 | 0.0478 | 0.0531 | 0.0500 |
| Satellite 2 | P68(m) | 0.0880 | 0.2323 | 0.0747 | 0.0843 | 0.0743 |
| Satellite 2 | P95(m) | 0.2912 | 0.4833 | 0.3013 | 0.2609 | 0.2714 |

**Table 4. Five independent runs 95% confidence interval (Root Mean Square Error).**

| Satellite | Fixed weighting | Huber-M | 3σ gate | Methodology of this paper |
|---|---|---|---|---|
| Satellite 1 | [0.2459,0.2482] | [0.1226,0.1259] | [0.1183,0.1213] | [0.1177,0.1210] |
| Satellite 2 | [0.2459,0.2482] | [0.1226,0.1259] | [0.1183,0.1213] | [0.1177,0.1210] |

demonstrating that the proposed method's weights (red curve) exhibit a strong negative correlation with variance (adapting continuously to real-time data quality), whereas the robust baselines only reduce weights at extreme variance peaks and fixed weighting remains constant. The bottom subplot further details the robust baselines' weighting characteristics: while both Huber-M (pink dotted) and 3σ weighting (cyan dotted) lower weights at gross error locations (circular markers), they fail to adjust for moderate variance regions—highlighting the proposed method's unique advantage in optimizing for both extreme outliers and non-extreme noise. Together, these results confirm the proposed dynamic weighting method's superior adaptability and error suppression capabilities, particularly for low-speed rotating GNSS scenarios with non-stationary noise and intermittent gross errors.

Table 3 quantifies the positioning performance of the proposed method against the original pseudorange error, fixed-weight filtering, Huber-M estimation, and 3σ gate-weighted filtering across two satellites (PRN1 and PRN2). For PRN1, the proposed method achieves an RMSE of 0.1476 m, which is comparable to the 3σ gate-weighted baseline (0.1452 m) and superior to fixed-weight filtering (0.2976 m) and the original error (0.2345 m), while its MAE (0.1017 m) is slightly better than Huber-M (0.1000 m) and 3σ weighting (0.1021 m). In terms of percentile errors, the proposed method's P50 (0.0644 m), P68 (0.1219 m), and P95 (0.3460 m) demonstrate a balance between middle-error concentration, 1σ-range precision, and 95%-reliability bounds, outperforming fixed-weight filtering by a significant margin and competing closely with the robust baselines. For PRN2, the proposed method exhibits even stronger advantages: an RMSE of 0.1150 m (lower than Huber-M's 0.1202 m and 3σ weighting's 0.1157 m), an MAE of 0.0778 m (the lowest among all methods), and percentile errors (P50 = 0.0500 m, P68 = 0.0743 m, P95 = 0.2714 m) that consistently outperform fixed-weight filtering and

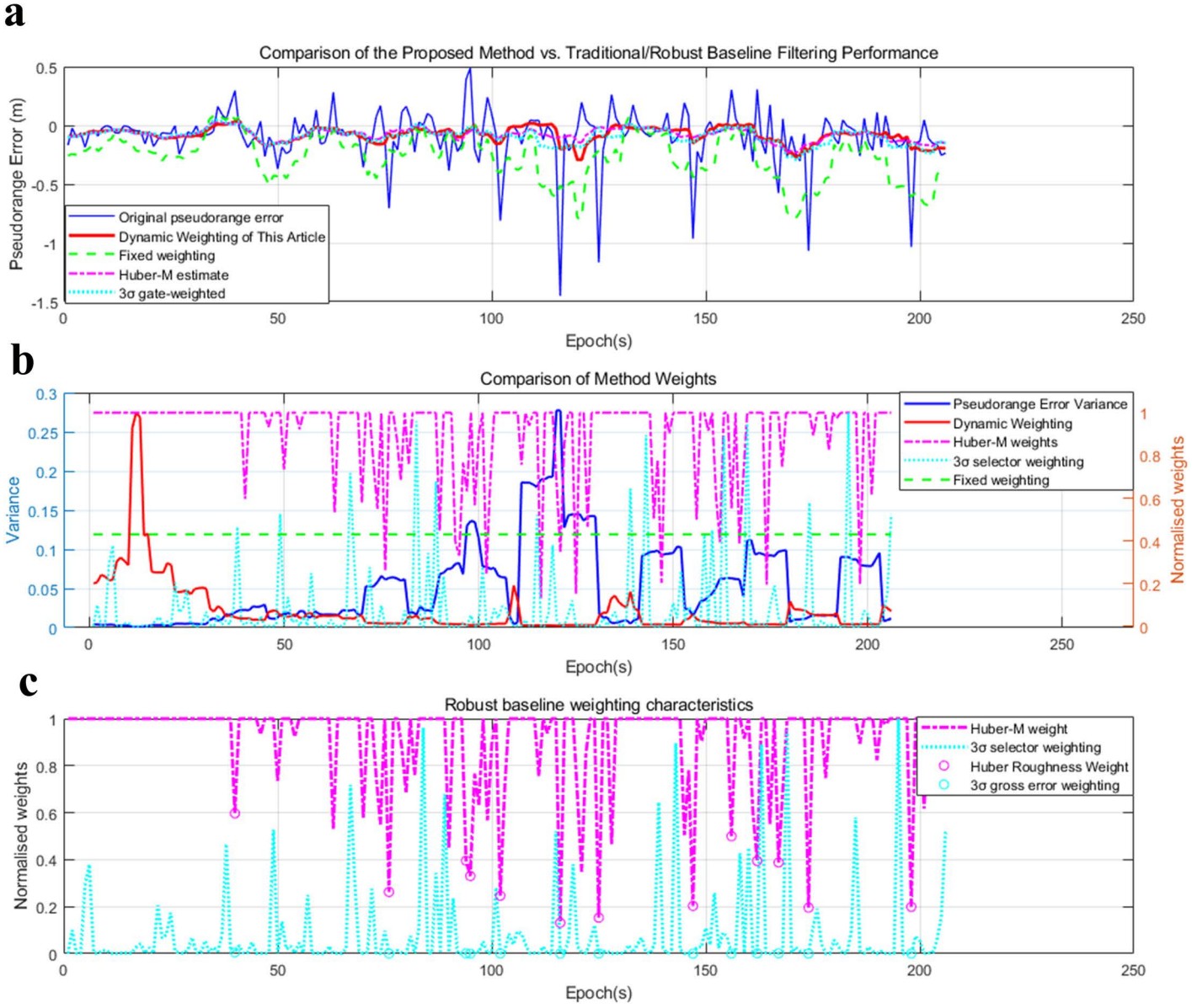

**Fig 10. Robustness Verification Experiment Diagram.** (a) Comparison of the Proposed Method vs. Traditional/Robust Baseline Filtering Performance. (b) Comparison of Method Weights. (c) Robust baseline weighting characteristics[90°_20240717110227_003.dat, filtering_baseline.m].

match or exceed the robust baselines. Collectively, these metrics confirm that the proposed dynamic weighting method delivers competitive or superior accuracy across all evaluated error metrics, with particular strengths in reducing extreme errors (evident in P95 improvements over fixed-weight filtering by ~45% for PRN1 and ~44% for PRN2) and maintaining precision in both moderate and high-reliability scenarios[90°_20240717110227_003.dat, filtering_baseline.m].

Table 4 presents the 95% confidence intervals (CIs) of RMSE across five independent runs for fixed-weight filtering, Huber-M estimation, 3σ gate-weighted filtering, and the proposed method, evaluated on satellites PRN1 and PRN2 (with identical results for both PRNs). For PRN1 and PRN2, the proposed method exhibits the narrowest CI for RMSE ([0.1177, 0.1210] m), closely followed by the 3σ gate-weighted baseline ([0.1183, 0.1213] m), indicating comparable stability but

marginally superior precision. Huber-M estimation shows a slightly wider and higher CI ([0.1226, 0.1259] m), while fixed-weight filtering yields the largest and highest CI ([0.2459, 0.2482] m)—more than double the RMSE range of the proposed method. Notably, the CI of the proposed method does not overlap with those of Huber-M or fixed-weight filtering, confirming statistically significant improvements in accuracy. The consistency between PRN1 and PRN2 further validates the robustness of these findings. Collectively, these results demonstrate that the proposed method achieves not only the lowest RMSE but also stable performance across repeated trials, outperforming both traditional and robust baselines in statistical reliability[90°_20240717110227_003.dat, filtering_baseline.m].

Traditional filtration algorithms usually remove observation and prediction data when detecting a significant deviation from the predicted value to avoid impacting positioning accuracy, reducing the available satellites for the system. However, the proposed method adjusts the participation of the satellites in the positioning solution through weight allocation. Compared to the traditional method, the current approach enhances the geometric distribution of localized satellites by maximizing the retention of those included in the filtering algorithm. For instance, the simulated results when considering a speed of 1/4 r/s are illustrated as follows.

In Fig 11, the x-axis represents the epoch, while the y-axis represents the number of satellites available for localization. The blue line represents the results obtained using the proposed method, while the red line represents the results obtained using the original method. When utilizing the proposed weighted filtering algorithm at a rate of 1/4 r/s, satellites that were originally dismissed as outliers but exhibited a tolerable level of error can be repurposed to enhance positioning accuracy[satnum.m, aReplayData_RC_4spr.dat].

In Fig 12 the horizontal coordinate represents the epoch, while the vertical coordinate represents the PDOP value. The blue line depicts the results of the proposed method, while the red line depicts the results of the original method. Comparing both methods under the case of 1/4 r/s illustrates that the PDOP value of the proposed method is lower than that of

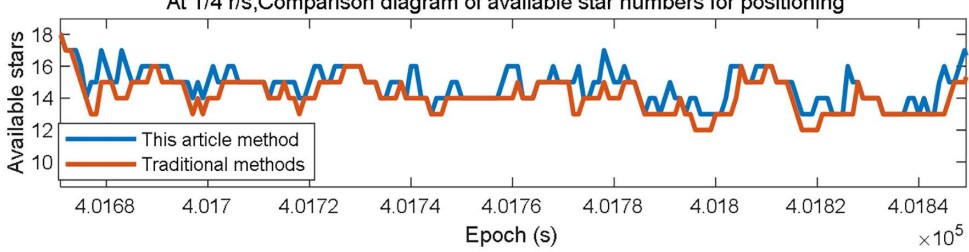

**Fig 11. Comparison of the quantity of stars accessible for localization at 1/4 r/s between the presented algorithm and the conventional algorithm.**

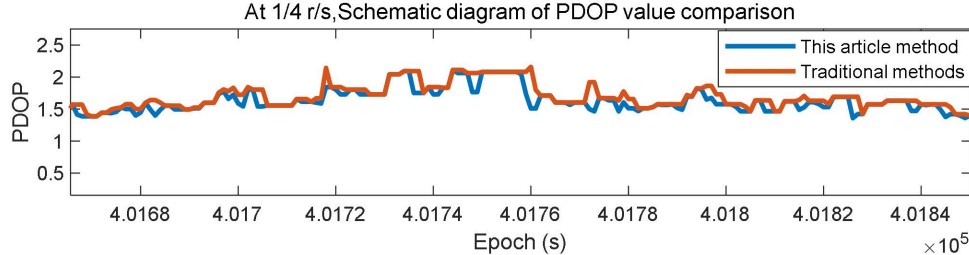

**Fig 12. Comparison of PDOP values between the present algorithm and the conventional algorithm at 1/4 r/s.**

the traditional method overall, resulting in better localization accuracy[aNavData-New-pdop-4spr.csv, aNavData-Tradition-pdop-4spr.csv, satnum.m, aReplayData_RC_4spr.dat].

## 5. Conclusions

In this research paper, the weighted positioning algorithm based on Chebyshev polynomials is deeply explored and analyzed. The behavior of satellite signal reception during low-speed rotation combined with the influence of the installation method and number of rotating carrier antennas on the satellite navigation positioning performance is analyzed, and it is proposed that under the premise of satisfying the receiving star coverage capability, the use of as few antennas as possible should be given priority, i.e., two antennas is the optimal solution. In addition, a Kalman filter algorithm based on independent model identification of observation weights is proposed. The algorithm uses Chebyshev extrapolation algorithm to judge the range of data accuracy, formulate reasonable weight function rules, and classify data with different accuracy. The receiver state equation based on the GNSS positioning system is established, the receiver state is predicted based on the state transfer matrix, and the mean square error of the a priori estimate, observation residual, and measurement variance are calculated. Chebyshev polynomials are used to extrapolate the pseudorange value, and the Chebyshev coefficient matrix is determined by the least squares method to provide the weight setting for the positioning solution. According to the extrapolated pseudo-distance value, the equivalent weight function is set, and reasonable weights are assigned to the data of different states, and then the new a posteriori estimation error of the mean square error array and measurement variance are obtained, so as to realize the accurate measurement update of the state estimation value. Then the test platform is designed, with the help of the dual-antenna rotating carrier test platform and application environment, and the dual-antenna positioning technology is adopted to ensure that at least one of the antennas can receive satellite signals in the rotating process, to expand the range of the received signals, and to improve the success rate of satellite signal capture. The traditional filtering algorithm and the weighted positioning algorithm with Chebyshev polynomials are comprehensively tested, including the test of elevation positioning accuracy, the test of horizontal error, the test of the number of available satellites and the test of PDOP value. By comparing the experimental results of the two algorithms under different rotational speeds, the weighted positioning algorithm with Chebyshev polynomials visually demonstrates the significant advantages of the Chebyshev polynomials in terms of positioning accuracy, stability of error, utilization of the number of available satellites, and optimization of the PDOP value, and thus fully verifies the feasibility and effectiveness of the algorithm.

Although the algorithm in this paper has improved the positioning accuracy under the condition of low rotational speed, the actual application scenarios are complicated and diverse, such as urban high-rise buildings, mountainous terrain and other environments, the satellite signals are easily blocked and reflected, resulting in the multipath effect, and the algorithm does not fully take into account the impact of these complex environmental factors on the positioning accuracy, and the positioning performance under such environments needs to be further researched and verified. In-depth study of the propagation characteristics of satellite signals in complex environments (e.g., urban canyons, mountainous areas, indoor areas, etc.) is needed to establish a more accurate signal model, incorporate the multipath effect and signal obstruction into the algorithm, and further optimize the weighted positioning algorithm based on Chebyshev polynomials, so as to improve the accuracy and reliability of the algorithm, and to enable it to be better applied to practical scenarios, such as urban navigation and emergency rescue. The algorithm is based on Chebyshev polynomials.

## Supporting information

**S1 Data.** S1 File. 90°_20240717110227_003.dat. Data from robustness validation experiments. S2 File. NavSolveKalman.c. Weighted Kalman Filter Algorithm in C Code. S3 File. NavSolveKalman.h. Weighted Kalman Filter Algorithm in C Code. S4 File. Chebshev.c. Chebyshev Polynomial C Code. S5 File. Chebshev.h. Chebyshev Polynomial C Code. S6 File. Antenna Gain Data.xls. Antenna gain data. S7 File. Antenna.m. Antenna gain pattern map MATLAB code. S8 File.

WeightSet.m. Weight Fitting Matlab Code. S9 File. aNavData-New-high-4spr.csv. Elevation data obtained using the paper methods at a rotational speed of 1/4 r/s. S10 File. aNavData-New-high-12spr.csv. Elevation data obtained using the paper methods at a rotational speed of 1/12 r/s. S11 File. aNavData-New-high-18spr.csv. Elevation data obtained using the paper methods at a rotational speed of 1/18 r/s. S12 File. aNavData-Tradition-high-4spr.csv. Elevation data obtained using traditional methods at a rotational speed of 1/4 r/s. S13 File. aNavData-Tradition-high-12spr.csv. Elevation data obtained using traditional methods at a rotational speed of 1/12 r/s. S14 File. aNavData-Tradition-high-18spr.csv. Elevation data obtained using traditional methods at a rotational speed of 1/18 r/s. S15 File. aReplayData_RC_4spr.dat. Positioning data acquired at a rotational speed of 1/4 revolutions per second. S16 File. aReplayData_RC_12spr.dat. Positioning data acquired at a rotational speed of 1/12 revolutions per second. S17 File. aReplayData_RC_18spr.dat. Positioning data acquired at a rotational speed of 1/18 revolutions per second. S18 File. func_XYZtoBLH.m. Matlab code for converting positioning data from three-dimensional coordinates to latitude and longitude. S19 File. Nav_CalcBlhHoriErr.m. Latitude and Longitude High-Speed Rotation Horizontal Error MATLAB Code. S20 File. ShuiP.m. Horizontal error output results Matlab code. S21 File. filtering_baseline.m. Robustness Verification Experiment Code. S22 File. satnum.m. Matlab code for PDOP value comparison. S23 File. aNavData-New-pdop-4spr.csv. Data on PDOP values for the proposed method. S24 File. aNavData-Tradition-pdop-4spr.csv. Data on PDOP values for the conventional method.
(ZIP)

## Author contributions

**Conceptualization:** Jiahui Gan, Peng Wu, Lu Feng.

**Data curation:** Jiahui Gan, ZhiChun Dai, Rundong Li.

**Formal analysis:** Peng Wu, ZhiChun Dai, Rundong Li.

**Funding acquisition:** Peng Wu.

**Investigation:** Peng Wu, ZhiChun Dai, Rundong Li.

**Methodology:** Peng Wu.

**Project administration:** Jiahui Gan, Peng Wu, Lu Feng.

**Resources:** Lu Feng.

**Software:** Jiahui Gan.

**Supervision:** Peng Wu, Lu Feng.

**Validation:** Jiahui Gan, Lu Feng.

**Visualization:** Lu Feng, ZhiChun Dai.

**Writing – original draft:** Jiahui Gan.

**Writing – review & editing:** Jiahui Gan, Peng Wu, Lu Feng, Rundong Li.

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
