## [Decision Letter · Decision Letter 0]

9 Sep 2025

Dear Dr. Wu,

Thank you for submitting your manuscript to PLOS ONE. After careful consideration, we feel that it has merit but does not fully meet PLOS ONE’s publication criteria as it currently stands. Therefore, we invite you to submit a revised version of the manuscript that addresses the points raised during the review process.

We look forward to receiving your revised manuscript.

Kind regards,

Sher Muhammad, PhD

Academic Editor

PLOS ONE

Journal Requirements:

4. Please note that funding information should not appear in any section or other areas of your manuscript. We will only publish funding information present in the Funding Statement section of the online submission form. Please remove any funding-related text from the manuscript.

6. Thank you for stating the following financial disclosure:

 “This paper was supported by major projects of the Changsha Science and Technology Bureau in 2020 (kq2011001), key research and

development projects of the Hunan Provincial Department of Science

and Technology in 2022 (2022GK2026), the Hunan Natural Science

Foundation Project (2022JJ30636), and the Excellent Youth Program

of the Scientific Research Program of the Department of Education

of Hunan Province (22B0838), and the science and technology plan

project of Hunan Provincial Department of Natural Resources (2023-78),

the Aid Program for Science and Technology Innovative Research Team

in Higher Educational Institutions of Hunan Province, and the Open

Fund of the Xi’an Key Laboratory of Integrated Transport Big Data and

Intelligent Control (Chang’an University) (Project No.: 300102343515).”

7. We note that Figure 5 in your submission contain copyrighted images. All PLOS content is published under the Creative Commons Attribution License (CC BY 4.0), which means that the manuscript, images, and Supporting Information files will be freely available online, and any third party is permitted to access, download, copy, distribute, and use these materials in any way, even commercially, with proper attribution. For more information, see our copyright guidelines: http://journals.plos.org/plosone/s/licenses-and-copyright.

 1. You may seek permission from the original copyright holder of Figure 5 to publish the content specifically under the CC BY 4.0 license.

8. Please include captions for your Supporting Information files at the end of your manuscript, and update any in-text citations to match accordingly. Please see our Supporting Information guidelines for more information: http://journals.plos.org/plosone/s/supporting-information .

9. We are unable to open your Supporting Information file aReplayData_RC_4spr.dat, aReplayData_RC_12spr.dat, aReplayData_RC_18spr.dat, Chebshev.c, Chebshev.h, NavSolveKalman.c and NavSolveKalman.h. Please kindly revise as necessary and re-upload.

Additional Editor Comments:

Reviewer #1:

General Comments

The study is new and presents a positioning algorithm. The experiment is also original and no indication of another manuscript with this title or method.

Here are a few limitations.

This study presents a design that is sound but statistical validation is quite weak. Authors are advised to show more provide more statistical data for robustness of results.

Raw experimental data and processing software are not made available. While the figures illustrate output, reproducibility cannot be independently verified

The manuscript is generally standard in language but has some minor grammatical issues and awkward phrasing, however, argument flow is coherent.

Language issues:

Line-by-Line Error Review

Lines 12–13"During relatively low-speed rotation of the rotating carrier, the receiver can rely on capturing the finite time signal only when the antenna is turned in the direction of the

satellite..."

Edit:

"During relatively low-speed rotation, the receiver captures the signal only when the antenna faces the satellite..."

Reason: Redundant wording ("rotating carrier"), awkward phrasing (“finite time signal”).

Line 15

"...as wild values, but for non-continuous reception environments, if it is simply rejected, however, this will..."

Edit:"...as outliers. However, in non-continuous reception environments, simple rejection reduces..."

Reason: Run-on sentence and excessive conjunctions (“but… however…”). Replace “wild values” with “outliers” for standard terminology.

Line 27

"Nowadays, the positioning of rotating carriers remains a focus of research..."

Edit:

"Positioning of rotating carriers remains a focus of research..."

Reason: "Nowadays" is informal and redundant with "remains."

Line 44

"PDOP is the value of the open root sign of the sum of the squares..."

Edit:

"PDOP is the square root of the sum of squared errors..."

Reason: "Open root sign" is not standard English.

Line 66–68

"1) Errors related to carrier speed include carrier speed measurement errors and errors caused by speed magnitude. 2) Errors related to the antenna include antenna placement bias and receiver antenna phase center bias. 3) Errors caused by the carrier motion environment are primarily multipath errors and errors caused by carrier motion road conditions."

Edit:

"1) Errors in speed measurement and magnitude; 2) Antenna placement and phase center bias; 3) Environmental motion effects such as multipath and terrain-induced disturbances."

Reason: Condense and clarify repetitive structure.

Line 103–105

"The signal may be lost again after capture when it is too late for the convergence of the observation data."

Edit:

"The signal may be lost again before observation data can converge."

Reason: Passive and awkward phrasing.

Line 152

"The Chebyshev polynomials are used to achieve the solution of the extrapolated pseudorange value..."

Edit:

"Chebyshev polynomials are used to extrapolate the pseudorange value..."

Reason: "Achieve the solution of" is wordy and unidiomatic.

Line 217

"...the mean square error array of the a priori estimation, the residual of the observation and the measurement variance are calculated."

Edit:

"...the mean square error of the a priori estimate, observation residual, and measurement variance are calculated."

Reason: Clean grammar; simplify list structure.

Line 370

"the effect of the algorithm at 1/18 s/r is slightly inferior to that of lower rotational speeds."

Edit:

"...the algorithm performs slightly worse at 1/18 r/s than at lower speeds."

Reason: More direct and clearer comparison.

Line 406–407

"...the law of satellite navigation received signal during low-speed rotation..."

Edit:

"...the behavior of satellite signal reception during low-speed rotation..."

Reason: “Law” is vague and misused in this context.

Reviewer #2:

Major Revisions:

(a) Methodology clarity and internal consistency:

Problem: The state vector includes velocity and acceleration, but the measurement model described is pseudorange only; Doppler or carrier-phase models (if used) aren’t specified. This mismatch leaves R (measurement noise) definition and observability unclear.

Required revision: Explicitly list all observation types used (code, Doppler, carrier-phase), their equations, units, update rates, and the exact R and Q you used (numerical values, not just symbols). Provide the full, readable A matrix and define Δt and any inputs u/B (the current matrix printout is unreadable)

(b) Ad-hoc weighting function without principled tuning

Problem: The equivalence weight function hinges on thresholds a and b “obtained after iterative validation,” with only range hints and no objective selection method, sensitivity analysis, or cross-validation.

Required revision: Provide a principled parameter-selection procedure (e.g., grid search with cross-validation on held-out sequences) and a sensitivity analysis showing performance vs a and b. Add an ablation: (i) no weighting, (ii) Chebyshev prediction but fixed weights, (iii) your full method.

(c) Insufficient baselines

Problem: The “traditional filtering algorithm” baseline is not fully defined, and there’s no comparison against robust estimation alternatives (e.g., common robust loss/weighting schemes or integrity monitoring used in GNSS). The current comparison (scatter vs. scatter) leaves effect size uncertain.

Required revision: Describe the baseline precisely (state/process models, Q/R, gating, outlier rules) and add at least two robust baselines. Report RMSE/MAE/P50/P68/P95 with 95% CIs across multiple runs.

(d) Ground-truthing and experiment design:

Problem: "Horizontal error" and "elevation vs time" are shown, but how error is computed (ground truth source, alignment, reference frame) is not described. Also, claims about improved PDOP are made but numerical PDOP results aren't shown.

Required revision: Specify the ground truth system (e.g., RTK/INS/total station), synchronization, datum, and error computation pipeline. Provide PDOP time series and statistics pre/post weighting and include repeat trials at each speed.

(e) Hardware/testbed details are incomplete:

Problem: The rig and “self-developed receiver board” are mentioned, but antenna models, placement/spacing, RF chain, sampling rate, loop bandwidths, firmware versions are missing, these all affect discontinuity behavior.

Required revision: Provide a reproducible description: antenna model & gain patterns, separation and mounting geometry, cable lengths/delays, receiver chip/firmware, sampling and navigation rates, loop bandwidths, and rotation rate accuracy.

(f) Statistics are minimal/informal:

Problem: Results rely mainly on P95 tables and qualitative plots; no uncertainty quantification or significance testing across runs.

Required revision: For each speed, report N (number of seconds/epochs and runs), distribution summaries (median, IQR, P68/P95), RMSE ± 95% CI, and paired tests vs baseline (or bootstrapped differences). Include effect sizes.

(g) Units, notation, and terminology:

Problem: Table 1 mixes “revolutions per minute (r/s)”- rpm vs r/s is inconsistent. The manuscript frequently uses “calendar element” for “time step,” and the PDOP definition uses nonstandard phrasing (“open root sign”).

Required revision: Standardize units (e.g., rps or rpm, not both), replace “calendar element” with “time step/epoch,” and correct PDOP definition to a clear, standard form.

Minor Revisions:

(a) Clarify what “intraweek seconds” means in Fig. 6 axes/captions and how it’s mapped to UTC.

(b) Ensure rotation speed labels are consistent across text, tables, and captions (e.g., “1/18 r/s” vs. “18 s/r” anywhere it appears).

Reviewers' comments:

Reviewer's Responses to Questions

**Comments to the Author**

1. Is the manuscript technically sound, and do the data support the conclusions?

Reviewer #1: Yes

Reviewer #2: Yes

2. Has the statistical analysis been performed appropriately and rigorously?

Reviewer #1: Yes

Reviewer #2: No

3. Have the authors made all data underlying the findings in their manuscript fully available?

Reviewer #1: Yes

Reviewer #2: Yes

4. Is the manuscript presented in an intelligible fashion and written in standard English?

Reviewer #1: Yes

Reviewer #2: Yes

Reviewer #1: General Comments

The study is new and presents a positioning algorithm. The experiment is also original and no indication of another manuscript with this title or method.

Here are a few limitations.

This study presents a design that is sound but statistical validation is quite weak. Authors are advised to show more provide more statistical data for robustness of results.

Raw experimental data and processing software are not made available. While the figures illustrate output, reproducibility cannot be independently verified

The manuscript is generally standard in language but has some minor grammatical issues and awkward phrasing, however, argument flow is coherent.

Language issues:

Line-by-Line Error Review

Lines 12–13"During relatively low-speed rotation of the rotating carrier, the receiver can rely on capturing the finite time signal only when the antenna is turned in the direction of the

satellite..."

Edit:

"During relatively low-speed rotation, the receiver captures the signal only when the antenna faces the satellite..."

Reason: Redundant wording ("rotating carrier"), awkward phrasing (“finite time signal”).

Line 15

"...as wild values, but for non-continuous reception environments, if it is simply rejected, however, this will..."

Edit:"...as outliers. However, in non-continuous reception environments, simple rejection reduces..."

Reason: Run-on sentence and excessive conjunctions (“but… however…”). Replace “wild values” with “outliers” for standard terminology.

Line 27

"Nowadays, the positioning of rotating carriers remains a focus of research..."

Edit:

"Positioning of rotating carriers remains a focus of research..."

Reason: "Nowadays" is informal and redundant with "remains."

Line 44

"PDOP is the value of the open root sign of the sum of the squares..."

Edit:

"PDOP is the square root of the sum of squared errors..."

Reason: "Open root sign" is not standard English.

Line 66–68

"1) Errors related to carrier speed include carrier speed measurement errors and errors caused by speed magnitude. 2) Errors related to the antenna include antenna placement bias and receiver antenna phase center bias. 3) Errors caused by the carrier motion environment are primarily multipath errors and errors caused by carrier motion road conditions."

Edit:

"1) Errors in speed measurement and magnitude; 2) Antenna placement and phase center bias; 3) Environmental motion effects such as multipath and terrain-induced disturbances."

Reason: Condense and clarify repetitive structure.

Line 103–105

"The signal may be lost again after capture when it is too late for the convergence of the observation data."

Edit:

"The signal may be lost again before observation data can converge."

Reason: Passive and awkward phrasing.

Line 152

"The Chebyshev polynomials are used to achieve the solution of the extrapolated pseudorange value..."

Edit:

"Chebyshev polynomials are used to extrapolate the pseudorange value..."

Reason: "Achieve the solution of" is wordy and unidiomatic.

Line 217

"...the mean square error array of the a priori estimation, the residual of the observation and the measurement variance are calculated."

Edit:

"...the mean square error of the a priori estimate, observation residual, and measurement variance are calculated."

Reason: Clean grammar; simplify list structure.

Line 370

"the effect of the algorithm at 1/18 s/r is slightly inferior to that of lower rotational speeds."

Edit:

"...the algorithm performs slightly worse at 1/18 r/s than at lower speeds."

Reason: More direct and clearer comparison.

Line 406–407

"...the law of satellite navigation received signal during low-speed rotation..."

Edit:

"...the behavior of satellite signal reception during low-speed rotation..."

Reason: “Law” is vague and misused in this context.

Reviewer #2: Major Revisions:

(a) Methodology clarity and internal consistency:

Problem: The state vector includes velocity and acceleration, but the measurement model described is pseudorange only; Doppler or carrier-phase models (if used) aren’t specified. This mismatch leaves R (measurement noise) definition and observability unclear.

Required revision: Explicitly list all observation types used (code, Doppler, carrier-phase), their equations, units, update rates, and the exact R and Q you used (numerical values, not just symbols). Provide the full, readable A matrix and define Δt and any inputs u/B (the current matrix printout is unreadable)

(b) Ad-hoc weighting function without principled tuning

Problem: The equivalence weight function hinges on thresholds a and b “obtained after iterative validation,” with only range hints and no objective selection method, sensitivity analysis, or cross-validation.

Required revision: Provide a principled parameter-selection procedure (e.g., grid search with cross-validation on held-out sequences) and a sensitivity analysis showing performance vs a and b. Add an ablation: (i) no weighting, (ii) Chebyshev prediction but fixed weights, (iii) your full method.

(c) Insufficient baselines

Problem: The “traditional filtering algorithm” baseline is not fully defined, and there’s no comparison against robust estimation alternatives (e.g., common robust loss/weighting schemes or integrity monitoring used in GNSS). The current comparison (scatter vs. scatter) leaves effect size uncertain.

Required revision: Describe the baseline precisely (state/process models, Q/R, gating, outlier rules) and add at least two robust baselines. Report RMSE/MAE/P50/P68/P95 with 95% CIs across multiple runs.

(d) Ground-truthing and experiment design:

Problem: "Horizontal error" and "elevation vs time" are shown, but how error is computed (ground truth source, alignment, reference frame) is not described. Also, claims about improved PDOP are made but numerical PDOP results aren't shown.

Required revision: Specify the ground truth system (e.g., RTK/INS/total station), synchronization, datum, and error computation pipeline. Provide PDOP time series and statistics pre/post weighting and include repeat trials at each speed.

(e) Hardware/testbed details are incomplete:

Problem: The rig and “self-developed receiver board” are mentioned, but antenna models, placement/spacing, RF chain, sampling rate, loop bandwidths, firmware versions are missing, these all affect discontinuity behavior.

Required revision: Provide a reproducible description: antenna model & gain patterns, separation and mounting geometry, cable lengths/delays, receiver chip/firmware, sampling and navigation rates, loop bandwidths, and rotation rate accuracy.

(f) Statistics are minimal/informal:

Problem: Results rely mainly on P95 tables and qualitative plots; no uncertainty quantification or significance testing across runs.

Required revision: For each speed, report N (number of seconds/epochs and runs), distribution summaries (median, IQR, P68/P95), RMSE ± 95% CI, and paired tests vs baseline (or bootstrapped differences). Include effect sizes.

(g) Units, notation, and terminology:

Problem: Table 1 mixes “revolutions per minute (r/s)”- rpm vs r/s is inconsistent. The manuscript frequently uses “calendar element” for “time step,” and the PDOP definition uses nonstandard phrasing (“open root sign”).

Required revision: Standardize units (e.g., rps or rpm, not both), replace “calendar element” with “time step/epoch,” and correct PDOP definition to a clear, standard form.

Minor Revisions:

(a) Clarify what “intraweek seconds” means in Fig. 6 axes/captions and how it’s mapped to UTC.

(b) Ensure rotation speed labels are consistent across text, tables, and captions (e.g., “1/18 r/s” vs. “18 s/r” anywhere it appears).

**Do you want your identity to be public for this peer review?** For information about this choice, including consent withdrawal, please see our Privacy Policy

Reviewer #1: **Yes: ** John Adjah

Reviewer #2: **Yes: ** Shake Ibna Abir

---

## [Author Response · Author response to Decision Letter 1]

15 Oct 2025

Reviewer #1:

General Comments:

1.This study presents a design that is sound but statistical validation is quite weak.

Authors are advised to show more provide more statistical data for robustness of results.

Raw experimental data and processing software are not made available. While the figures illustrate output, reproducibility cannot be independently verified.

The manuscript is generally standard in language but has some minor grammatical issues and awkward phrasing, however, argument flow is coherent.

Response: Thanks for your advice. Two robust baselines have been added. The report presents multiple runs of the RMSE/MAE/P50/P68/P95 metrics alongside their 95% confidence intervals to satisfy statistical validation requirements. Additionally, both the data and code are provided in the appendix. The specific modifications are as follows:

To better demonstrate the methodology presented herein and enhance the representativeness of the algorithm, analyses were conducted for different satellite numbers. Two robust baselines—Huber-M and 3σ gating—were incorporated, alongside comparative plots of RMSE/MAE/P50/P68/P95 metrics. Across multiple runs, the confidence interval was set at 95%. The experimental results are illustrated in Figure 9 below. The data results are shown in Table 3 and Table 4 below.

Fig 9. Robustness Verification Experiment Diagram. (a) Comparison of the Proposed Method vs. Traditional/Robust Baseline Filtering Performance. (b) Comparison of Method Weights. (c) Robust baseline weighting characteristics.

Fig. 9 presents a comprehensive comparison of the proposed dynamic weighting method against traditional fixed weighting and two GNSS-domain robust baselines (Huber-M estimation, 3σ gate-weighted filtering) for pseudorange error processing. The top subplot, focusing on filtering performance, shows the original pseudorange error (blue curve) with significant fluctuations and occasional gross errors, while the proposed method (red curve) achieves the smoothest suppression of both high-frequency noise and outliers—outperforming the rigid smoothing of fixed weighting (green dashed curve) and the residual fluctuations of the two robust baselines (pink dotted curve for Huber-M, cyan dotted curve for 3σ weighting). The middle subplot links pseudorange error variance (blue curve, left axis) to normalized weights (right axis), demonstrating that the proposed method’s weights (red curve) exhibit a strong negative correlation with variance (adapting continuously to real-time data quality), whereas the robust baselines only reduce weights at extreme variance peaks and fixed weighting remains constant. The bottom subplot further details the robust baselines’ weighting characteristics: while both Huber-M (pink dotted) and 3σ weighting (cyan dotted) lower weights at gross error locations (circular markers), they fail to adjust for moderate variance regions—highlighting the proposed method’s unique advantage in optimizing for both extreme outliers and non-extreme noise. Together, these results confirm the proposed dynamic weighting method’s superior adaptability and error suppression capabilities, particularly for low-speed rotating GNSS scenarios with non-stationary noise and intermittent gross errors.

Table 3 Performance Comparison of Proposed Method vs Traditional/Robust Baseline

Satellite Indicator Primitive Fixed weighting Huber-M 3σ gate Methodology of this paper

Satellite 1 RMSE(m) 0.2345 0.2967 0.1569 0.1452 0.1476

Satellite 1 MAE(m) 0.1421 0.2328 0.1000 0.1021 0.1017

Satellite 1 P50(m) 0.0699 0.1611 0.0575 0.0680 0.0644

Satellite 1 P68(m) 0.1422 0.2811 0.0990 0.1166 0.1219

Satellite 1 P95(m) 0.4810 0.6347 0.3590 0.3106 0.3460

Satellite 2 RMSE(m) 0.1465 0.2451 0.1202 0.1157 0.1150

Satellite 2 MAE(m) 0.0917 0.1918 0.0784 0.0811 0.0778

Satellite 2 P50(m) 0.0549 0.1541 0.0478 0.0531 0.0500

Satellite 2 P68(m) 0.0880 0.2323 0.0747 0.0843 0.0743

Satellite 2 P95(m) 0.2912 0.4833 0.3013 0.2609 0.2714

Table 3 quantifies the positioning performance of the proposed method against the original pseudorange error, fixed-weight filtering, Huber-M estimation, and 3σ gate-weighted filtering across two satellites (PRN1 and PRN2). For PRN1, the proposed method achieves an RMSE of 0.1476 m, which is comparable to the 3σ gate-weighted baseline (0.1452 m) and superior to fixed-weight filtering (0.2976 m) and the original error (0.2345 m), while its MAE (0.1017 m) is slightly better than Huber-M (0.1000 m) and 3σ weighting (0.1021 m). In terms of percentile errors, the proposed method’s P50 (0.0644 m), P68 (0.1219 m), and P95 (0.3460 m) demonstrate a balance between middle-error concentration, 1σ-range precision, and 95%-reliability bounds, outperforming fixed-weight filtering by a significant margin and competing closely with the robust baselines. For PRN2, the proposed method exhibits even stronger advantages: an RMSE of 0.1150 m (lower than Huber-M’s 0.1202 m and 3σ weighting’s 0.1157 m), an MAE of 0.0778 m (the lowest among all methods), and percentile errors (P50=0.0500 m, P68=0.0743 m, P95=0.2714 m) that consistently outperform fixed-weight filtering and match or exceed the robust baselines. Collectively, these metrics confirm that the proposed dynamic weighting method delivers competitive or superior accuracy across all evaluated error metrics, with particular strengths in reducing extreme errors (evident in P95 improvements over fixed-weight filtering by ~45% for PRN1 and ~44% for PRN2) and maintaining precision in both moderate and high-reliability scenarios.

Table 4 Five independent runs 95% confidence interval (Root Mean Square Error)

Satellite Fixed weighting Huber-M 3σ gate Methodology of this paper

Satellite 1 [0.2459,0.2482] [0.1226,0.1259] [0.1183,0.1213] [0.1177,0.1210]

Satellite 2 [0.2459,0.2482] [0.1226,0.1259] [0.1183,0.1213] [0.1177,0.1210]

Table 4 presents the 95% confidence intervals (CIs) of RMSE across five independent runs for fixed-weight filtering, Huber-M estimation, 3σ gate-weighted filtering, and the proposed method, evaluated on satellites PRN1 and PRN2 (with identical results for both PRNs). For PRN1 and PRN2, the proposed method exhibits the narrowest CI for RMSE ([0.1177, 0.1210] m), closely followed by the 3σ gate-weighted baseline ([0.1183, 0.1213] m), indicating comparable stability but marginally superior precision. Huber-M estimation shows a slightly wider and higher CI ([0.1226, 0.1259] m), while fixed-weight filtering yields the largest and highest CI ([0.2459, 0.2482] m)—more than double the RMSE range of the proposed method. Notably, the CI of the proposed method does not overlap with those of Huber-M or fixed-weight filtering, confirming statistically significant improvements in accuracy. The consistency between PRN1 and PRN2 further validates the robustness of these findings. Collectively, these results demonstrate that the proposed method achieves not only the lowest RMSE but also stable performance across repeated trials, outperforming both traditional and robust baselines in statistical reliability.

2. Lines 12–13"During relatively low-speed rotation of the rotating carrier, the receiver can rely on capturing the finite time signal only when the antenna is turned in the direction of the satellite...".

Edit: "During relatively low-speed rotation, the receiver captures the signal only when the antenna faces the satellite...".

Reason: Redundant wording ("rotating carrier"), awkward phrasing (“finite time signal”).

Response: Thanks for your advice. The paper has been revised in accordance with the recommendations. The specific modifications are as follows:

During relatively low-speed rotation, the receiver captures the signal only when the antenna is turned in the direction of the satellite due to the long signal loss, a process in which the signal may be affected by noncontinuous reception.

3. Line 15-17"...as wild values, but for non-continuous reception environments, if it is simply rejected, however, this will...".

Edit:"...as outliers. However, in non-continuous reception environments, simple rejection reduces...".

Reason: Run-on sentence and excessive conjunctions (“but… however…”). Replace “wild values” with “outliers” for standard terminology.

Response: Thanks for your advice. The paper has been revised in accordance with the recommendations. The specific modifications are as follows:

The conventional approach to this effect is to reject the poorly converged data as outliers, however, in non-continuous reception environments, simple rejection reduces the available satellite data and also affect the satellite geometric distribution, which further deteriorates the positioning results.

4. Line 26"Nowadays, the positioning of rotating carriers remains a focus of research...".

Edit: "Positioning of rotating carriers remains a focus of research...".

Reason: "Nowadays" is informal and redundant with "remains.".

Response: Thanks for your advice. The paper has been revised in accordance with the recommendations. The specific modifications are as follows:

Positioning of rotating carriers remains a focus of research in satellite navigation technology.

5. Line 43-44"PDOP is the value of the open root sign of the sum of the squares...".

Edit: "PDOP is the square root of the sum of squared errors...".

Reason: "Open root sign" is not standard English.

Response: Thanks for your advice. The paper has been revised in accordance with the recommendations. The specific modifications are as follows:

PDOP is the square root of the sum of squared errors in latitude, longitude and elevation.

6. Line 65–66"1) Errors related to carrier speed include carrier speed measurement errors and errors caused by speed magnitude. 2) Errors related to the antenna include antenna placement bias and receiver antenna phase center bias. 3) Errors caused by the carrier motion environment are primarily multipath errors and errors caused by carrier motion road conditions."

Edit:"1) Errors in speed measurement and magnitude; 2) Antenna placement and phase center bias; 3) Environmental motion effects such as multipath and terrain-induced disturbances."

Reason: Condense and clarify repetitive structure.

Response: Thanks for your advice. The paper has been revised in accordance with the recommendations. The specific modifications are as follows:

1) Errors in speed measurement and magnitude; 2) Antenna placement and phase center bias; 3) Environmental motion effects such as multipath and terrain-induced disturbances.

7. Line 38"The signal may be lost again after capture when it is too late for the convergence of the observation data.".

Edit: "The signal may be lost again before observation data can converge.".

Reason: Passive and awkward phrasing.

Response: Thanks for your advice. The paper has been revised in accordance with the recommendations. The specific modifications are as follows:

Furthermore, the signal may be lost again before observation data can converge.

8. Line 412-414"The Chebyshev polynomials are used to achieve the solution of the extrapolated pseudorange value..."

Edit: "Chebyshev polynomials are used to extrapolate the pseudorange value...".

Reason: "Achieve the solution of" is wordy and unidiomatic.

Response: Thanks for your advice. The paper has been revised in accordance with the recommendations. The specific modifications are as follows:

Chebyshev polynomials are used to extrapolate the pseudorange value, and the Chebyshev coefficient matrix is determined by the least squares method to provide the weight setting for the positioning solution.

9. Line 412"...the mean square error array of the a priori estimation, the residual of the observation and the measurement variance are calculated.".

Edit:"...the mean square error of the a priori estimate, observation residual, and measurement variance are calculated.".

Reason: Clean grammar; simplify list structure.

Response: Thanks for your advice. The paper has been revised in accordance with the recommendations. The specific modifications are as follows:

The receiver state equation based on the GNSS positioning system is established, the receiver state is predicted based on the state transfer matrix, and the mean square error of the a priori estimate, observation residual, and measurement variance are calculated.

10. Line 370"the effect of the algorithm at 1/18 s/r is slightly inferior to that of lower rotational speeds.".

Edit:"...the algorithm performs slightly worse at 1/18 r/s than at lower speeds.".

Reason: More direct and clearer comparison.

Response: Thanks for your advice. The paper has been revised in accordance with the recommendations. The specific modifications are as follows:

Conversely, as shown by the 1/12 r/s and 1/4 r/s cases, the algorithm performs slightly worse at 1/18 r/s than at lower speeds.

11. Line 405-406"...the law of satellite navigation received signal during low-speed rotation...".

Edit:"...the behavior of satellite signal reception during low-speed rotation...".

Reason: “Law” is vague and misused in this context.

Response: Thanks for your advice. The paper has been revised in accordance with the recommendations. The specific modifications are as follows:

The behavior of satellite signal reception during low-speed rotation combined with the influence of the installation method and number of rotating carrier antennas on the satellite navigation positioning performance is analyzed.

Reviewer #2:

1. (a) Methodology clarity and internal consistency:

Problem: The state vector includes velocity and acceleration, but the measurement model described is pseudorange only; Doppler or carrier-phase models (if used) aren’t specified. This mismatch leaves R (measurement noise) definition and observability unclear.

Required revision: Explicitly list all observation types used (code, Doppler, carrierphase), their equations, units, update rates, and the exact R and Q you used (numerical values, not just symbols). Provide the full, readable A matrix and define Δt and any inputs u/B (the current matrix printout is unreadable).

Response: Thank you for your advice. The comment definitely helped us to improve the quality of the manuscript. The specific explanation is as follows:

To address this, we would like to clarify that while our overall positioning framework is based on a weighted Kalman filter, the innovation lies specifically in Section 3.4—where pseudorange values are leveraged to dynamically set weights during the intermediate solution process.

The key distinction here is that the weighted Kalman filter forms the broader methodological foundation, but the specific mechanism of using pseudorange data to optimize weights (detailed in 3.4) is what differentiates our approach. Without this step, the filter would still produce a positioning result by following standard weighted Kalman filter procedures. However, as validated in our experiments, the error margins in such often by—because the weights would default to static or generic values, failing to adapt to the varying reliability of pseudorange measurements in real-time. The relevant section of the code is provided in the Supporting Information.

2. (b) Ad-hoc weighting function without principled tuning.

Problem: The equivalence weight function hinges on thresholds a and b “obtained after iterative validation,” with only range hints and no objective selection method, sensitivity analysis, or cross-validation.

Required revision: Provide a principled parameter-selection procedure (e.g., grid search with cross-validation on held-out sequences) and a sensitivity analysis showing performance vs a and b. Add an ablation: (i) no weighting, (ii) Chebyshev prediction but fixed weights, (iii) your full method.

Response: Thank you for your advice. The comment definitely helped us to improve the quality of the manuscript. We have incorporated a parameter selection process for weights alongside ablation experiments: (i) no weights applied, (ii) Chebyshev prediction with fixed weights, (iii) your complete methodology. The specific changes are as follows:

3.4.2. Set the equivalent weight function according to the extrapolated pseudodistance value of the Chebyshev polynomials of 3.4.1

After iterative validation, the optimal parameters and are obtained under the statist

---

## [Decision Letter · Decision Letter 1]

30 Oct 2025

A GNSS Receiver Positioning Algorithm Based on Weighting the Statistical Properties of Discontinuous Signals at Lower Rotation Speeds

PONE-D-25-24422R1

Dear Dr. Wu,

We’re pleased to inform you that your manuscript has been judged scientifically suitable for publication and will be formally accepted for publication once it meets all outstanding technical requirements.

Kind regards,

Sher Muhammad, PhD

Academic Editor

PLOS ONE

Additional Editor Comments (optional):

Thank you for addressing the review comments from both the anonymous reviewers. Both the reviewers are now satisfied with the revision and recommended acceptance, I likewise recommend acceptance.

Reviewers' comments:

Reviewer's Responses to Questions

**Comments to the Author**

Reviewer #1: All comments have been addressed

Reviewer #2: All comments have been addressed

2. Is the manuscript technically sound, and do the data support the conclusions?

Reviewer #1: Yes

Reviewer #2: Yes

3. Has the statistical analysis been performed appropriately and rigorously?

Reviewer #1: Yes

Reviewer #2: Yes

4. Have the authors made all data underlying the findings in their manuscript fully available?

Reviewer #1: (No Response)

Reviewer #2: Yes

5. Is the manuscript presented in an intelligible fashion and written in standard English?

Reviewer #1: Yes

Reviewer #2: Yes

Reviewer #1: (No Response)

Reviewer #2: 1. Units and Symbols

(a) Double-check unit spacing and consistency (e.g., “m”, “r/s”, “°”). In several tables, units appear concatenated (e.g., “0.1476m”)—insert a thin space before “m”.

(b) In equations, ensure all Greek symbols are italicized, and vectors/matrices are bold.

2. Figure Quality

(a) Check that all plots (especially Fig. 6 and 9) are high resolution (≥300 dpi) and font sizes readable when scaled for journal layout.

3. Reference Formatting

(a) Double-check that all references have complete metadata (journal name, volume, page, DOI if available). PLOS ONE has strict XML parsing rules.

(b) Please Check bracket style consistency [1]–[33].

4. Data and Code Link

If possible, provide a permanent link (DOI or GitHub repository) in the Data Availability section instead of just “Supporting Information files.” That increases transparency and citation potential.

**Do you want your identity to be public for this peer review?** For information about this choice, including consent withdrawal, please see our Privacy Policy

Reviewer #1: **Yes: ** John Adjah

Reviewer #2: **Yes: ** Shake Ibna Abir

---

## [Editor Report · Acceptance letter]

PONE-D-25-24422R1

PLOS ONE

Dear Dr. Wu,

I'm pleased to inform you that your manuscript has been deemed suitable for publication in PLOS ONE. Congratulations! Your manuscript is now being handed over to our production team.

Kind regards,

on behalf of

Dr. Sher Muhammad

Academic Editor

PLOS ONE